# Mechanistic analysis of multiple processes controlling solar-driven $H_2O_2$ synthesis using engineered polymeric carbon nitride

Yubao Zhao [1,2✉], Peng Zhang [1], Zhenchun Yang[1], Lina Li[1], Jingyu Gao [1], Sheng Chen [3], Tengfeng Xie[4], Caozheng Diao [5], Shibo Xi [5], Beibei Xiao[6], Chun Hu[1] & Wonyong Choi [2✉]

Solar-driven hydrogen peroxide ($H_2O_2$) production presents unique merits of sustainability and environmental friendliness. Herein, efficient solar-driven $H_2O_2$ production through dioxygen reduction is achieved by employing polymeric carbon nitride framework with sodium cyanaminate moiety, affording a $H_2O_2$ production rate of 18.7 $\mu$mol h$^{-1}$ mg$^{-1}$ and an apparent quantum yield of 27.6% at 380 nm. The overall photocatalytic transformation process is systematically analyzed, and some previously unknown structural features and interactions are substantiated via experimental and theoretical methods. The structural features of cyanamino group and pyridinic nitrogen-coordinated soidum in the framework promote photon absorption, alter the energy landscape of the framework and improve charge separation efficiency, enhance surface adsorption of dioxygen, and create selective $2e^-$ oxygen reduction reaction surface-active sites. Particularly, an electronic coupling interaction between $O_2$ and surface, which boosts the population and prolongs the lifetime of the active shallow-trapped electrons, is experimentally substantiated.

[1] Key Laboratory for Water Quality and Conservation of the Pearl River Delta, Ministry of Education & Institute of Environmental Research at Greater Bay, Guangzhou University, Guangzhou, P. R. China. [2] Division of Environmental Science and Engineering, Pohang University of Science and Technology (POSTECH), Pohang, Korea. [3] Key Laboratory for Soft Chemistry and Functional Materials, Ministry of Education, Nanjing University of Science and Technology, Nanjing, P. R. China. [4] College of Chemistry, Jilin University, Changchun, P. R. China. [5] Singapore Synchrotron Light Source, National University of Singapore, Singapore, Singapore. [6] School of Energy and Power Engineering, Jiangsu University of Science and Technology, Zhenjiang, P. R. China. ✉email: ybzhao@gzhu.edu.cn; wchoi@postech.edu

Hydrogen peroxide ($H_2O_2$) is a versatile chemical, functioning as a green oxidant and a clean liquid fuel. In this context, photochemical $H_2O_2$ production is attracting great interests as an alternative solar fuel option[1-5]. Solar-driven $H_2O_2$ synthesis presents unique features of remarkable sustainability and environmental friendliness, as compared to the traditional anthraquinone process and direct synthesis method[6-18]. However, the gap between the potentials of solar energy and practical application of it is the cost-effectiveness. Moreover, the earth-abundant photocatalyst composition and high efficiency of photons to chemical conversion process are two critical factors for cost-effectiveness[19,20]. Therefore, design of a low-cost photocatalyst with decent solar conversion efficiency is the major challenge to the goal of sustainable $H_2O_2$ production. It is particularly worth noting that the efficiency improvement relies heavily on the comprehensive mechanistic understanding of the structure–activity relationship in nanoscale[20].

A plethora of photocatalysts with earth-abundant elements have been developed for efficient $H_2O_2$ production. Among various photocatalysts, polymeric carbon nitride (PCN) consisting of the organic framework has the advantage of easy structural optimization[21-27]; its surface dangling functional groups and the conjugated electronic structure can be facilely modified for efficient catalytic reactions. This unique advantage can be maximized only if the structure–photocatalytic activity relationship is clearly understood and taken into account in designing the PCN structure and composition[28-30]. However, understanding the overall photocatalytic process is challenging since it involves multiple consecutive steps, which include photon absorption/excitation, emissive decay, photo-induced charge trapping/separation, charge transport to the surface-active-sites, interfacial charge transfer with the surface adsorbed reactants, intermediates conversions, and finally product desorption (Fig. 1). Each step in the complicated process contributes to the overall solar conversion efficiency[31,32].

The pendant amino group on PCN is proven to introduce energetically deeper trapping sites which may negatively influence the photocatalytic activity[32]. Various methods for modifying the structure and composition of PCN have been investigated to improve its photocatalytic activity[33,34]. For example, substitution of the pendant amino groups by cyanamide units improves the solar hydrogen evolution reaction (HER) activity of the PCN photocatalysts[29,30,35]. However, a comprehensive mechanistic understanding on how the specific structural features influence each step in the photocatalytic $2e^-$ oxygen reduction reaction (ORR) is challenging and unexplored, while such information is critical for the rational design of a highly efficient solar-driven $H_2O_2$ production system. Herein, superior solar-driven $2e^-$ ORR performance is achieved on an engineered PCN framework. The key mechanistic features of the overall photocatalytic process are analyzed by tracing the consecutive electrons transfer steps involved in the photoinduced processes and the subsequent surface reactions. Introduction of sodium cyanaminate moiety by molten-salt treatment creates the electron-withdrawing cyanamino-goup and coordinative interaction between sodium and pyridinic nitrogen, and thus alters the distribution of the energy landscape of the carbon nitride framework. The structural evolution improves photon absorption capacity and retardes emissive charge-recombination by trapping a significant fraction of charge carriers. It also leads to enhanced accumulation of surface charges, stronger interaction beween surface and dioxygen, and higher density of catalytic active sites for selective $2e^-$ ORR.

## Results

### Synthesis and $H_2O_2$ production performance of the photocatalysts.
PCN was synthesized by condensation reaction of the melamine-cyanuric acid complex under high temperature (Fig. 2a)[36]. The resulting PCN was further treated with sodium thiocyanate (NaSCN) molten salt to tailor the conjugated electronic structure and the surface properties. Further condensation reaction occurs in the molten salt and leads to two favorable structural features: (1) improved polymerization degree and expanded conjugated electronic structure[37-40]; and (2) conversion of the amino group to the sodium cyanaminate moiety (Fig. 2a and Fig. S1 and S2). The PCN framework with sodium cyanaminate moiety is denoted by PCN-NaCA; and PCN-NaCA-$n$ ($n$ = 1, 2, and 3) refers to the sample with specified salt/PCN weight ratio (0.5, 1, and 2). Molten salt treated samples are similar in morphology (Fig. S3) but different in BET surface area and porosity (Fig. S4 and Table S1). By thermal gravimetric analysis-gas chromatography mass spectrometry (TGA-GC-MS), the major components of the evolved gases during synthesis were identified (Figs. S5–S7); and the mechanism of the conversion of amino group to cyanaminate moiety was proposed accordingly (Fig. S8)[41].

The sodium content of PCN-NaCA-2 was determined to be 6.9 wt.% by inductively coupled plasma atomic emission spectroscopy (ICP-OES). For further elucidating the chemical environment of sodium in the framework, PCN-NaCA framework was simulated with density functional theory (DFT) based on a linear melon structure with infinite repeating units; and in the optimized configuration, the interaction between the sodium ion and four pyridinic nitrogen atoms of two adjacent heptazine units is observed (Fig. 2b and Fig. S9). Nitrogen K-edge x-ray near-edge structure (XANES) measurements were thereafter used to investigate the interaction (Fig. 2c). PCN shows typical $\pi^*$ resonance at 399.3 eV corresponding to pyridinic N of the tri-s-triazine moiety[42,43]; most importantly, blue shift and enhancement of the pyridinic N peak is observed, demonstrating the presence of coordination interaction and charge transfer from pyridinic nitrogen to sodium[44,45]. Meanwhile, there is no shift for the other $\pi^*$ resonances (401.2 eV, amino; 402.3 eV, graphitic)[46,47], indicating no interaction between these nitrogen atoms and sodium. Furthermore, the chemical environment of the sodium was characterized with X-ray absorption spectrometer (Fig. S10). Sodium in PCN-NaCA-2 presents a K-edge absorption profile that is different from that of NaSCN and NaSCN/PCN, but is very similar to the profile of sodium diformylamine. The XAS results are indicating that the sodium ion is strongly interacting with PCN-NaCA-2 framework via the coordination with pyridinic nitrogen, which matches well with the theoretical simulation.

The photocatalytic selective reduction of $O_2$ to $H_2O_2$, in an ideal scenario, should employ $H_2O$ as the proton/electron donor, so that there is no additional $CO_2$ emission from this process[48-51]. However, the electron and proton extraction through water oxidation process ($2H_2O \rightarrow 4H^+ + 4e^- + O_2$) is inefficient as the hole transfer kinetics toward water oxidation is sluggish, causing severe charge recombination, which results in low solar conversion efficiency[52]. In natural photosynthesis, electrons/protons are extracted from water via complicated bio-enzymatic reactions and subsequently used in transforming $CO_2$ to biomass to achieve the energy-uphill reaction[53,54]. An alternative solution is to utilize more reactive biomass derivatives (as an electron/proton donor instead of water) that are abundant and cheap. In the rapidly rising biodiesel industry, glycerol is the byproduct and its yield accounts for 10 wt% of the biodiesel production, but the limited consumption of glycerol makes it surplus[55]. Developing a proper process of consuming and valorizing glycerol well matches the market need. The crude glycerol is cheap with price of 10–15 c per lb. (80 wt%, Oleoline), while $H_2O_2$ is a moderately valuable chemical with price of around 6.0 USD per lb. (35 wt%, Supleco).

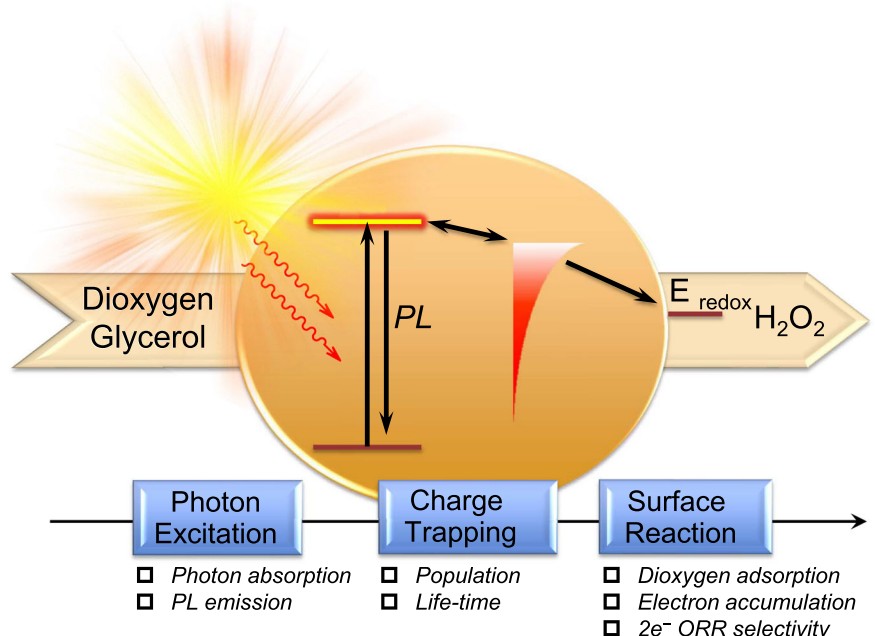

**Fig. 1** Illustration of the sequential mechanisms of the photophysical, photochemical, and surface reaction steps involved in solar-driven $H_2O_2$ production process.

Moreover, glycerol is non-toxic and bio-degradable, rendering it an ideal electron/proton donor[48–51]. Therefore, solar production of $H_2O_2$ with biodiesel-derived glycerol as the electron/proton donor can be proposed as an environmentally benign and cost-effective solution.

Figure 3a compares the photoproduction of $H_2O_2$ in the presence of glycerol using various photocatalysts in 50 mL batch reaction. PCN and PCN-400 (PCN heated in nitrogen atmosphere at 400 °C for 3 h) exhibit the same photocatalytic performance, generating 0.23 mM $H_2O_2$ in 45 min irradiation, while PCN-NaCA-n samples exhibit enhanced activity for $H_2O_2$ production. PCN-NaCA-2, with the lowest BET surface area of 11.9 $m^2\,g^{-1}$ among the photocatalysts (Table S1), shows the optimum photocatalytic performance, producing 2.80 mM $H_2O_2$ in 45 min irradiation in the presence of 3.5 wt% glycerol (photocatalyst and glycerol concentration are optimized, Figs. S11 and S12). The $H_2O_2$ production rate reaches 187 $\mu mol\,h^{-1}$ (by 10 mg PCN-NaCA-2) under solar simulator irradiation, which is superior to the most recently reported highly active photocatalytic systems (Table S2)[11,12,15–18]. Mesoporous carbon nitride (mpg-$C_3N_4$), which is an efficient photocatalyst for $H_2$ evolution[56], shows low performance in the selective $2e^-$ ORR, producing only 0.53 mM $H_2O_2$. As the $H_2O_2$ production reaction involves transfer of protons as well as electrons, the acidic media is normally more favorable[15–17]. However, PCN-NaCA-2 favors neutral to mild basic conditions for efficient solar-driven $H_2O_2$ production (Fig. 3b and Fig. S13). As dioxygen is the source for $H_2O_2$ production, the oxygen partial pressure directly impacts the $H_2O_2$ production performance; air atmosphere lowers the $H_2O_2$ production, and in the nitrogen atmosphere, trace $H_2O_2$ is produced from the residual dissolved oxygen in water after nitrogen flushing (Fig. S14).

The hydrogen peroxide decomposes slowly on PCN-NaCA-2 and PCN (Fig. S15), which contributes positively to the $H_2O_2$ accumulation in the reaction system. PCN-NaCA-2 shows stable performance for recycling, and at 8th run, the photocatalytic performance is 88.1% of the initial run (Fig. S16). Long-term running stability of PCN-NaCA-2 was examined in a batch

reaction, and $H_2O_2$ concentration reaches 11.1 mM after 10 h irradiation (Fig. S17). The characterizations of the recycled photocatalyst indicate that long-term photocatalytic reaction leads the loss of the amine moiety as well as sodium ion in the framework (Fig. S18).

The apparent quantum yield (AQY) of the photoproduction of $H_2O_2$ was measured using monochromatic light as a function of wavelength (Fig. 3c). The AQY is 27.6% and 11.8% at 380 nm and 420 nm, respectively, and rapidly decreases with further increasing the wavelength, which matches well with the absorption spectral profile of the photocatalyst. The fact that the action spectrum of $H_2O_2$ production is closely correlated with the optical absorption spectrum supports the photocatalytic mechanism based on ORR.

To further explore the performance of PCN-NaCA-2 in large scale reaction, the continuous serial micro-batch reactor, which typically has high surface-area-to-volume ratio and allows for more efficient irradiation of the solid-gas-liquid triphasic reaction system, is employed (Fig. S19)[57–60]. As shown in Fig. 3d, the $H_2O_2$ production rate increases with PCN-NaCA-2 concentration, and reaches plateau when concentration is higher than 2400 $mg\,L^{-1}$. With the photocatalyst concentration of 2400 $mg\,L^{-1}$, $H_2O_2$ concentration reaches high values of 12.3 and 18.6 mM with short retention time of 36 min and 72 min, respectively. While for PCN, flow photo-reaction with 36 min retention time affords only 0.5 mM $H_2O_2$ under the same reaction conditions. PCN-NaCA-2 shows a $H_2O_2$ production performance which is 24.6 times of that on PCN. The photocatalytically produced $H_2O_2$ aqueous solution is an enviornmental benign and efficient reductant for elimination of the highly-toxic hexavalent Cr (Fig. S20).

There are two pathways for $H_2O_2$ production, e.g., one-step two electrons transfer pathway and superoxide radicals involved pathway[61]. For analyzing the $H_2O_2$ formation mechanism, the contribution of each pathway in the reaction system was evaluated. As shown in the electron spin resonance (ESR) spectra (Fig. S21a), the peak intensity of DMPO (5,5-dimethyl-1-pyrroline N-oxide)−superoxide radical adduct, although being weak, increases with irradiation time, stating the photocatalytic

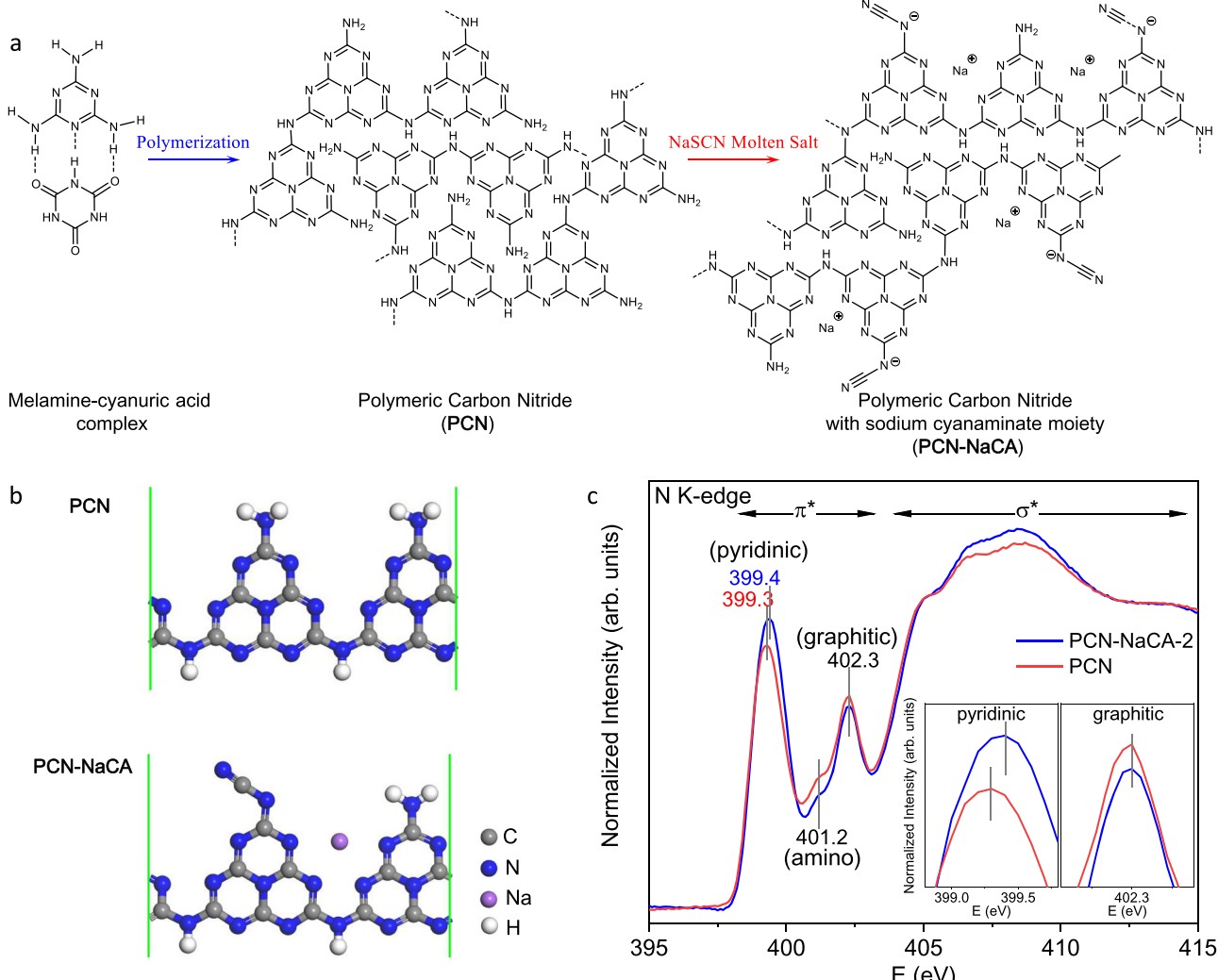

**Fig. 2 Synthesis and structure of the photocatalysts. a** Synthesis of polymeric carbon nitride (PCN) and polymeric carbon nitride with cyanaminate moiety (PCN-NaCA-$n$). **b** The optimized configurations of PCN and PCN-NaCA by DFT calculation. **c** N K-edge x-ray absorption near edge structure (XANES) of PCN and PCN-NaCA-2. The insets show enlarged peaks of pyridinic nitrogen and graphitic nitrogen.

production of superoxide radicals. Superoxide dismutase (SOD) can efficiently accelerate the superoxide radical dismutation reaction towards $H_2O_2$ production in both biological and photocatalytic reaction systems[62–64]. For evaluating the contribution of the superoxide involved pathway to the overall $H_2O_2$ production, SOD was added in PCN-NaCA-2 photocatalyzed reaction system. However, with the increase of SOD concentration, there is no obvious acceleration of the $H_2O_2$ production observed (Fig. S21b). This probe reaction is demonstrating that the superoxide radicals involved pathways contribute negligibly to the overall $H_2O_2$ production.

The biomass-derived glycerol serves as the electron/proton donor for $O_2$ to $H_2O_2$ conversion. The glycerol degration process was investigated via reaction intermediates identification. By gas chromatography-mass spectrometry (GC-MS), four major intermediates, e.g., dihydroxylacetone, glyceraldyhyde, glyceric acid, and glycolic acid, are identified. Based on these major intermediates, the glycerol degradation process by hole oxidation is proposed accordingly (Fig. S22). As there are multiple intermediates/products in glycerol oxidation reaction, selective oxidation of 4-methylbenzyl alcohol to 4-methylbenzaldehyde was employed for further examining the $H_2O_2$ selectivity. As shown in Fig. S23, $H_2O_2$ production selectivity (in terms of the

molar ratio beween $H_2O_2$ and 4-methylbenzaldehyde) reaches >93%, stating that the alcohol to aldehyde conversion is supplying electrons/protons to the dioxygen reduction for $H_2O_2$ production[65].

The distinction between the photocatalytic performances of PCN and PCN-NaCA-2 varies with the reaction conditions. In the flow-reactor under sufficient photons irradiation and with glycerol as the proton/electron donor, PCN-NaCA-2 presents optimum $H_2O_2$ production activity, which is 24.6 times of that on PCN. In the photocatalytic $H_2O_2$ production coupled with BPA (Bisphenol A) degradation, the initial $H_2O_2$ production rate and BPA degradation reaction rate constant on PCN-NaCA-2 are five times of those on PCN (Fig. S24); while in the anaerobic $H_2$ evolution reaction, PCN-NaCA-2 and PCN exhibit very close photocatalytic activity in terms of $H_2$ production rate (Fig. S25). It is proposed that PCN-NaCA-2 possesses unique surface-active sites for efficient outputting of the photoinduced electrons to surface adsorbed dioxygen, and its potential for $H_2O_2$ production can be maximized in the presence of sufficient supply of electron/proton (Fig. S25).

**Mechanistic investigations on the photons to chemical energy conversion process.** For a comprehensive understanding of the

rationale for the above-mentioned photocatalytic performance of the cyanaminate-modified PCN, we carried out systematic mechanistic investigations on the following aspects: (1) excitation and emissive decay process; (2) non-emissive states, focusing on population and decay kinetics of the trapped electrons; and (3) surface processes that include charge carriers diffusion behavior, dioxygen adsorption, and ORR activity and selectivity (Fig. 1).

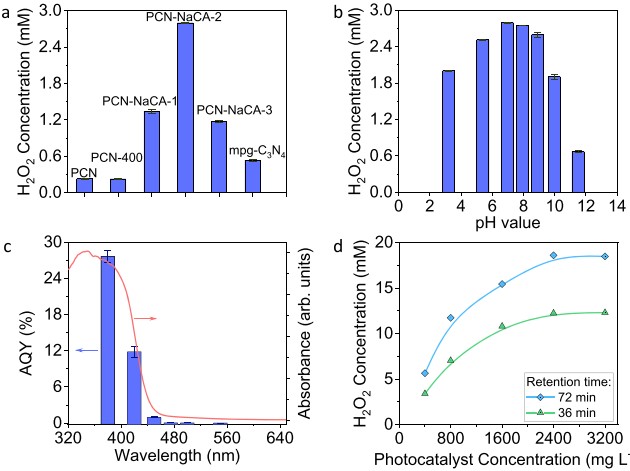

**Fig. 3 Solar-driven $H_2O_2$ production performance. a** Comparison of the performances of the photocatalysts in batch reactor with 45 min solar simulator irradiation; PCN, polymeric carbon nitride; PCN-400, PCN calcined at 400 °C for 3 h in $N_2$; PCN-NaCA-1— PCN-NaCA-3, polymeric carbon nitride with a various amount of sodium cyanaminate moieties; mpg-$C_3N_4$, mesoporous carbon nitride. **b** The pH-dependence of $H_2O_2$ production performance on PCN-NaCA-2. **c** The apparent quantum yield of $H_2O_2$ production as a function of wavelength on PCN-NaCA-2 (blue bars) and the UV–Vis diffuse reflectance spectra of PCN-NaCA-2 (red curve). **d** $H_2O_2$ production performance as a function of photocatalyst concentration on PCN-NaCA-2 in continuous flow photo-reactor. The error bars show the standard deviation from the mean value of $H_2O_2$ production (**a**, **b**) and AQY (**c**) from triplicate experiments.

As compared to PCN, the photon absorption spectra of PCN-NaCA-$n$ samples show improved absorbance at 350–380 nm, which is commonly observed in the conjugated aromatic systems with $\pi-\pi^*$ transition (Fig. 4a)[66,67]. The absorbance at 450–500 nm might result from the excitation to the defects states below the conduction band. In the Tauc plots from Kubelka–Munk function transformation, the optical band gap is determined to be 2.75 eV for PCN (Fig. 4b). Introduction of the sodium cyanaminate moiety into the carbon nitride framework narrows the band gap, e.g., 2.73, 2.69, and 2.63 eV for PCN-NaCA-1, PCN-NaCA-2, and PCN-NaCA-3, respectively. PCN and PCN-NaCA-2 show the same valence band potential of 1.57 V (vs. RHE) as determined by valence-band x-ray photoelectron spectra (Fig. 4c)[68]. The conduction band potentials of PCN and PCN-NaCA-2 are accordingly determined to be −1.18 and −1.12 V (vs. RHE), demonstrating that $2e^-$ ORR by conduction band electrons is thermodynamically feasible (Fig. 4d). The enhanced photon absorption is a primary prerequisite for the high activity of $H_2O_2$ production, as this step provides the initial driving force for the whole solar energy conversion process.

Upon photon absorption, the excited photocatalyst will either relax to the ground state via photoluminescence (PL) or transit to non-emissive state through charge trapping wherein some of the trapped charges will participate in the expected surface chemical reaction steps[69]. Steady-state and transient PL spectroscopy is thus employed as an indirect method for analyzing the situation of the trapped charges[70]. As shown in Fig. 4e, PCN shows strong PL emission peak at 488 nm, while the emission intensity of PCN-NaCA-2 is weaker. Moreover, considering the stronger photon absorption of PCN-NaCA-2 than PCN, much larger proportion of the excited states should transit to the non-emissive states on PCN-NaCA-2 than PCN at this stage. It is also interesting to note a blue shift of 18 nm for PCN-NaCA-2 as compared to that of PCN, which is attributed to the quantum confinement caused by the decreased thickness of the layer stacking[71–74].

To further understand the variation of the electronic structure of the conjugated system with sodium cyanaminate moiety, the decay kinetics of the emissive state is thereafter analyzed by time resolved photoluminescence spectroscopy. As shown in Fig. 4f,

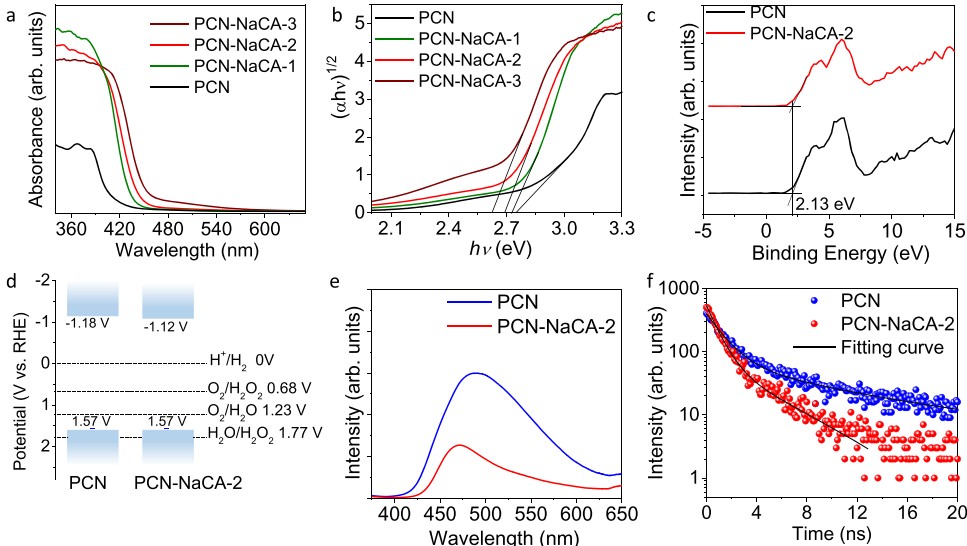

**Fig. 4 Excitation and quenching properties of the photocatalysts. a** UV–Vis diffuse reflectance spectra of the photocatalysts: PCN polymeric carbon nitride, PCN-NaCA-1–PCN-NaCA-3 polymeric carbon nitride with a various amount of sodium cyanaminate moieties. **b** Plots of transformed Kubelka–Munk function versus photon energy. Colors in **a** and **b**: PCN, black; PCN-NaCA-1, olive; PCN-NaCA-2, red; PCN-NaCA-3, wine. **c** Valence band X-ray photoelectron spectra of PCN (black) and PCN-NaCA-2 (red). **d** Band structure of PCN and PCN-NaCA-2. **e** Steady-state photoluminescence spectra of PCN (blue) and PCN-NaCA-2 (red) under 355 nm excitation. **f** Time-resolved photoluminescence (TRPL) spectra of PCN (blue) and PCN-NaCA-2 (red).

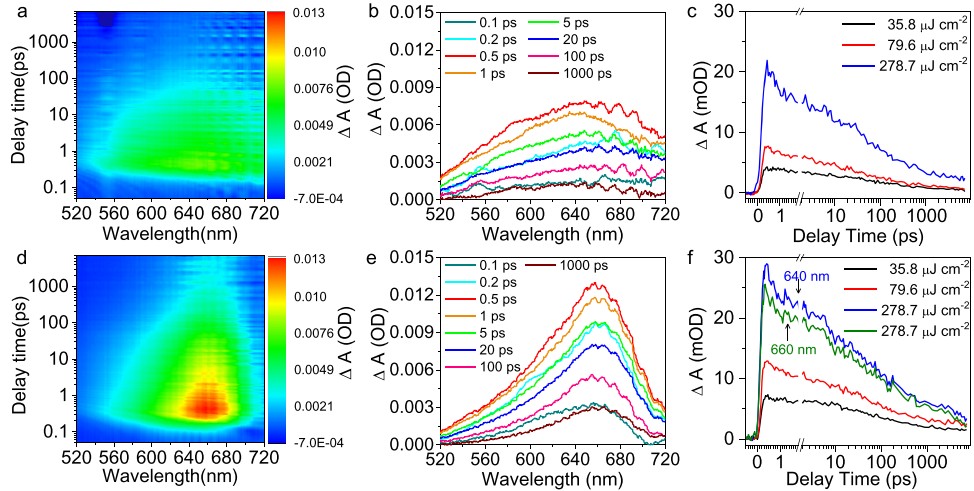

**Fig. 5 Femtosecond transient absorption spectra (fs-TAS) of PCN-NaCA-2. a, b** fs-TAS upto 7 ns delay times after excitation by 365 nm laser pulse with energy density of 79.6 μJ cm$^{-2}$ in glycerol aqueous solution (3.5 wt%) under vacuum. **c** fs-TAS decay kinetics profiles monitored at the wavelength of 640 nm under vacuum with a series of excitation energy densities. **d, e** fs-TAS upto 7 ns delay times after excitation by 365 nm laser pulse with energy density of 79.6 μJ cm$^{-2}$ in glycerol aqueous solution (3.5 wt%) in 1 atm. O$_2$ atmosphere. **f** fs-TAS decay kinetics profiles monitored at the wavelength of 640 and 660 nm in pure O$_2$ atmosphere with a series of excitation energy densities. PCN-NaCA-2, polymeric carbon nitride with sodium cyanaminate moiety.

PCN-NaCA-2 shows faster PL decay kinetics than that of PCN, showing average lifetime of 3.46 and 8.27 ns for PCN-NaCA-2 and PCN, respectively (Table S3). The shorter lifetime and weaker PL emission intensity of PCN-NaCA-2 compared to PCN indicates the fast quenching of luminescence. This might be attributed to the fact that charge separation is enhanced by extended π-conjugated systems and delocalization of the π-electrons due to improved polymerization degree[75,76].

The non-emissive trapped electrons have the potential for participating chemical conversion reaction on the surface. We then focus on the status of the non-emissive trapped electrons and their interaction with the surface adsorbed dioxygen on PCN and PCN-NaCA-2. Femtosecond transient absorption spectroscopy (fs-TAS) was thus employed to quantitatively monitor the population of the trapped electrons and their decay kinetics. The main objective in this stage is to elucidate the interaction between the adsorbed dioxygen and the photoinduced electrons. All the fs-TAS characterization was thus conducted in the presence of glycerol as the electron/proton donor.

Under vacuum condition, after excitation by laser pulse with photon fluence of 79.6 μJ cm$^{-2}$, PCN-NaCA-2 presents characteristic broad absorption peak at 640 nm, which is identified as trapped electrons (Fig. 5a, b and Fig. S26)[77]. Figure 5c shows the decay kinetics profiles of the photoinduced electrons with various photon fluence, and the initial absorption intensity depends on the photon fluence, demonstrating the direct impact of the photon fluence on the population of the trapped photoinduced electrons. With the increase of the photon fluence from 35.8 to 79.6 μJ cm$^{-2}$, the half-life time ($t_{0.5}$) of the trapped electrons decreases from 45 to 25 ps (Fig. 6). However, further increasing the excitation energy to 278.7 μJ cm$^{-2}$ slightly changed $t_{0.5}$ (24 ps), which indicates that the effect of excitation fluence on the electron life time is saturated under high photon fluence. In the presence of glycerol as the electron donor under vacuum, the photoinduced electrons accumulate and create electric filed, which accelerates the decay kinetics of the photo-induced electrons[77].

In lieu of vacuum condition, femtosecond-TAS was measured in pure O$_2$ atmosphere for monitoring the impact of the surface adsorbed dioxygen on the trapped electrons. Since dioxygen is an efficient electron acceptor, the accumulation of trapped electrons

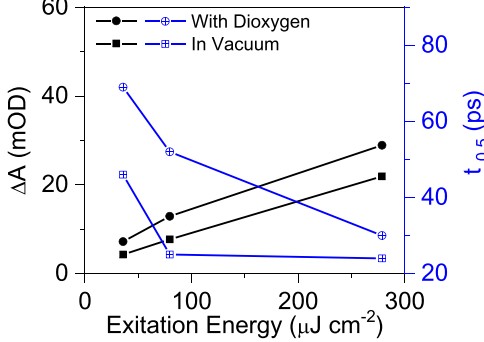

**Fig. 6 The fluence dependence of the initial amplitude (black lines and plots) and the decay half-life time (blue lines and plots) of the photo-induced electrons of PCN-NaCA-2 in glycerol aqueous solution in pure oxygen atmosphere (solid circle and plus center circle) and vacuum (solid square and plus center square) conditions.** PCN-NaCA-2, polymeric carbon nitride with sodium cyanaminate moiety.

(i.e., transient absorption intensity) is expected to be lower under oxygen atmosphere than under vacuum. Contrary to the expectation, the presence of dioxygen markedly enhances the transient absorption intensity, and modulates the shape of the absorption peak at 660 nm as compared with that in vacuum condition under the same photon fluence conditions (Fig. 5d–f and Fig. S26). The intense absorption peak indicates higher population of trapped electrons, and the well-defined shape of the absorption peak indicates that the distribution of the electron trapping species/sites is different from those in vacuum.

Upon the fs-laser excitation, the initial transient absorption peak increases with the excitation energy intensity, and the slope is similar between the vacuum and O$_2$ atmosphere conditions (Fig. 6 and Fig. S27). The fact that the transient absorption of trapped electrons is consistently higher in the presence of O$_2$ than in vacuum regardless of the excitation energy intensity implies that more trapped electrons are induced in the presence of surface adsorbed O$_2$. In addition, it should be also noted that the half-life time (in 10–100 ps range) of the trapped electrons in oxygen atmosphere is longer than that in vacuum. This indicates that the

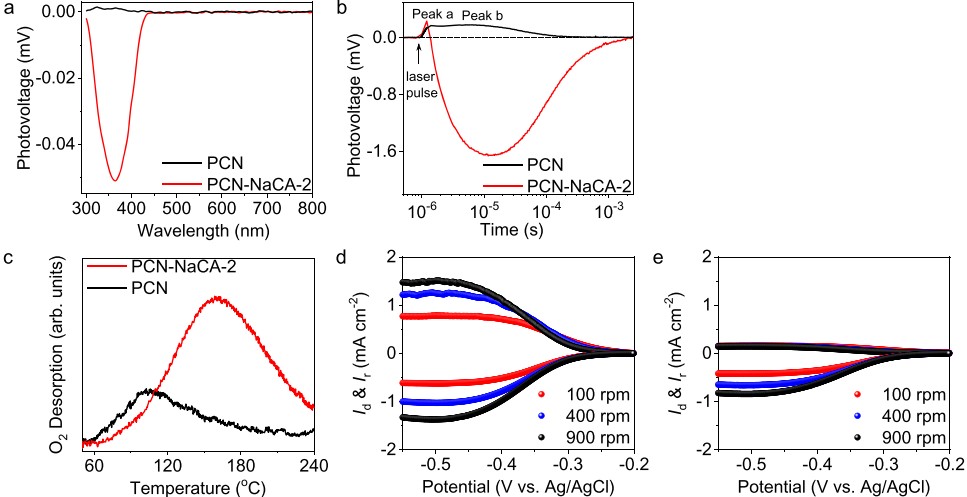

**Fig. 7 Behavior of the charges and dioxygen on the surfaces of PCN and PCN-NaCA-2. a** Surface photovoltage (SPV) spectra of PCN (black curve) and PCN-NaCA-2 (red curve). **b** The transient photovoltage spectra of PCN (black curve) and PCN-NaCA-2 (red curve) with 100 μJ 355 nm pulse excitation. **c** Temperature programmed oxygen desorption ($O_2$-TPD) profiles of PCN (black curve) and PCN-NaCA-2 (red curve). **d**, **e** The linear sweep voltammetry (LSV) plots of PCN-NaCA-2 (**d**) and PCN (**e**) on rotating ring disk electrode (RRDE); $I_d$ is the disk current density, and $I_r$ is the ring current density divided by collection efficiency. PCN polymeric carbon nitride, PCN-NaCA-2 polymeric carbon nitride with sodium cyanaminate moiety.

interaction between surface adsorbed dioxygen and the photo-induced electrons starts at a very early stage of electrons trapping step (in ps to ns time scale), which is opposite to the fact that the interfacial electron transfer from the irradiated semiconductor to $O_2$ occurs much later in μs to ms time scale[77]. It seems that the dioxygen adsorption induces the formation of electron trapping sites on the surface of PCN-NaCA-2 in the ps–ns time range and the transfer of trapped electrons to $O_2$ occurs at a much later stage (μs–ms time region). On the other hand, pure PCN exhibited no absorption peak of trapped electrons in fs-TAS, and only a bleaching signal at around 480–540 nm is observed (Fig. S28) in both $O_2$ atmosphere and vacuum conditions. There is obvious difference in the shape and position of the bleach signal, i.e., sharp peak at 494 nm under vacuum and broad peak at 510 nm in $O_2$ conditions. The decay kinetics of the bleach signal for PCN is, however, similar in vacuum and dioxygen atmosphere (Fig. S29). The clear effects of $O_2$ on the TAS profiles in PCN-NaCA-2 and PCN systems demonstrate that the dioxygen adsorption directly influences the electronic structures of the polymeric carbon nitride framework. The comparison of fs-TAS between PCN-NaCA-2 and PCN confirms that the photoinduced electron accumulation is uniquely observed on PCN-NaCA-2, not on PCN; and surface dioxygen adsorption further increases the population and prolongs the life time of the photo-induced electrons on PCN-NaCA-2. These characteristics of PCN-NaCA-2 should make it suitable for producing $H_2O_2$ through surface catalytic ORR process.

For initiating oxygen reduction reactions, the photoinduced charge carriers must diffuse to the surface active-sites. The amount of the photoinduced charges and the charges diffusion properties are of great importance, and thus be investigated by surface photovoltage (SPV) spectroscopy. The intensity of SPV signal is positively related with the amount of photoinduced charges as well as spatial charge separation properties; and the sign (positive or negative) of photovoltage is correlated to the direction of the charges diffusion[78,79]. As shown in Fig. 7a, PCN presents very weak positive photovoltage at the band of 300–350 nm; while PCN-NaCA-2 presents an intensive negative SPV signal at the band of 300–400 nm, demonstrating the enhancement of charge separation efficiency on PCN-NaCA-2 than that on PCN[80–83]. The steady-state observation here is consistent with

fs-TAS results, in which the amount of shallow trapped electron produced on PCN-NaCA-2 is higher than that on PCN.

The dynamics behavior of the charge carriers is further investigated by transient photovoltage (TPV) characterization. For both PCN and PCN-NaCA-2, there are two photovoltage peaks (peak a and peak b, Fig. 7b), which are, respectively, attributed to drift and diffusion of the photo-induced charges[84,85]. PCN-NaCA-2 presents a strong and negative peak b lasting for 2.5 ms before decaying to zero, speaking for the diffusion of large number of photoinduced electrons to the surface after photon excitation event. The photovoltage (PV) characterizations demonstrate, obviously, that the depletion of the charge carriers by recombination is attenuated by the construction of the sodium cyanaminate moiety on polymeric carbon nitride framework.

Electrochemical impedance spectroscopic analysis was conducted to further analysed the type of conductivity of PCN-NaCA-2 (Fig. S30). In the Mott–Schottky plots, the negative and positive slopes, respectively, correspond to p-type and n-type conductivities[86]. The p-type conductivity might be attributed to the strong electron withdrawing property of the cyanamino-group, and the coordinative interaction between sodium and pyridinic nitrogen in the framework[87]. These structural features could lead to a unique energy landscape that may promote the charge separation under illumination.

The final step in the photo-production of $H_2O_2$ is the interfacial transfer of the trapped electrons to the adsorbed $O_2$. The interaction between the catalysts surface and the dioxygen molecules is thus studied by the temperature programmed oxygen desorption ($O_2$-TPD), wherein the area of the desorption peak indicates the amount of dioxygen adsorbed per unit catalyst mass and the desorption temperature estimates the surface interaction strength of dioxygen (Fig. S31). As shown in Fig. 7c, PCN-NaCA-2 exhibits higher $O_2$ adsorption capacity (by mass), which is around three times larger than that of the pristine PCN. Moreover, PCN-NaCA-2 has a much lower BET surface area than that of PCN, e.g., 11.9 $m^2\,g^{-1}$ for PCN-NaCA-2 versus 83.2 $m^2\,g^{-1}$ for PCN. This implies that the density of the surface adsorbed dioxygen on PCN-NaCA-2 is around 20 times larger than that on PCN. More importantly, it is noted that PCN-NaCA-2 has stronger surface binding strength for $O_2$, as the $O_2$

desorption peaks on PCN-NaCA-2 and PCN appear at 160 °C and 104 °C, respectively. The stronger surface binding strength for $O_2$ as well as the high density of adsorbed $O_2$ should contribute positively to the enhanced ORR activity.

Finally, for an efficient $H_2O_2$ production, the selectivity towards $2e^-$ ORR pathway is of critical importance. The performance of $2e^-$ ORR to $H_2O_2$ is thus evaluated by analyzing its electrochemical selectivity on a rotating ring disc electrode (RRDE), wherein the disc current comes from the dioxygen reduction reactions and the ring current comes from the $2e^-$ oxidation of $H_2O_2$ produced from the disc. Figure 7d shows linear sweep voltammetry (LSV) curves with PCN-NaCA-2 as the active material in oxygen-saturated KOH electrolyte. It should be particularly noted that the disc current density and ring current density reach $-0.5677$ and $0.7507\,\text{mA cm}^{-2}$, respectively (at the applied voltage of $-0.425$ V (vs. Ag/AgCl) and rotation speed of 100 rpm), which yields a $H_2O_2$ selectivity of 99.8%. Under the same test condition, PCN exhibits much lower disc current density and ring current density of $-0.3838$ and $0.1547\,\text{mA cm}^{-2}$, respectively, which gives a $H_2O_2$ selectivity of 46.6% (Fig. 7e). The RRDE measurements thus demonstrate that PCN-NaCA-2 exhibits superior activity and selectivity for $2e^-$ ORR as compared to PCN. This indicates that the formation of sodium cyanaminate moiety on PCN-NaCA-2 creates surface active sites, which are particularly favorable for $2e^-$ ORR pathway.

It is difficult to further reveal the reaction mechanism by monitoring the transient intermediates on the surface via experimental methodologies. However, computational quantum chemistry offers a platform that makes the prediction of the surface-structure dependent reaction mechanism feasible[88]. Density functional theory (DFT) calculation of the ORR reaction mechanism (both $2e^-$ and $4e^-$ ORR pathways) is therefore performed for further understanding the rationale behind the superior performance of PCN-NaCA-2 for $H_2O_2$ production. After optimization (Figs. S32–S35 and Tables S4–S6), the stability of ORR intermediates (in terms of the adsorption energies) on PCN is generally higher than that on PCN-NaCA (Tables S7). The stronger adsorption strength of ORR intermediates on PCN can help to capture initial reactant, but may have a negative effect on the formation of final product. As shown in Fig. 8, the reduction of OOH* and OH* to form the final products of $H_2O_2$ and $H_2O$ are the rate-determining steps of $2e^-$ and $4e^-$ ORR processes. The $\Delta G$ values of the reduction of OOH* and OH* on PCN-NaCA are calculated to be 0.23 and 0.37 eV, respectively, both of which are much smaller than that on PCN with $\Delta G$ values

of 0.96 and 0.93 eV, indicating that PCN-NaCA can exhibit higher ORR catalytic activity. This could be the fundamental reason for the improved ORR catalytic activity on PCN-NaCA-2 in the electrochemical measurements. Furthermore, the reduction of OOH* and OH* are associated with the selectivity of $2e^-$ and $4e^-$ ORR pathways[89]. For PCN-NaCA, the $\Delta G$ of the reduction of OOH* (0.23 eV) is 0.14 eV lower than that of OH* reduction (0.37 eV), rendering much higher probability of $2e^-$ ORR pathway. On the contrary, $\Delta G$ of the endothermic OOH* reduction step (0.96 eV) on PCN is slightly higher (0.03 eV) than that of OH* reduction (0.93 eV), stating that $2e^-$ and $4e^-$ ORR pathways are both feasible with slightly higher probability of $4e^-$ ORR pathways. The theoretical observations here are highly consistent with the previously mentioned experimental RRDE results, in which the $2e^-$ ORR selectivity on PCN and PCN-NaCA-2 are, respectively, 46.6 and 99.8% at the applied voltage of $-0.425$ V (vs. Ag/AgCl) and rotation speed of 100 rpm.

## Discussion

In the solar-driven selective $2e^-$ ORR using the biomass-derived glycerol as the electron/proton donor, the carbon nitride framework with cyanaminate sodium salt moiety exhibits superior photoactivity for $H_2O_2$ production, which is 24.6 times of that on PCN in a continuous flow photoreaction. Introducing the sodium cyanaminate moiety to the PCN framework has the following multiple effects: (1) enhancing photon absorption, (2) altering the energy landscape of the framework and leading to retarded radiative charge recombination and improved electrons accumulation in the surface region, (3) constructing surface active sites for dioxygen adsorption, and (4) favoring selective $2e^-$ ORR, all of which synergically contributes to the performance of solar-driven $H_2O_2$ production. It is particularly worth-noting that the interaction between surface adsorbed dioxygen and PCN-NaCA-2 boosts the population, and prolongs the lifetime of the shallow-trapped electrons. This indicates that the reactants-surface interactions during photon excitation, along with the intrinsic excitation properties of the materials, should be taken into consideration for designing an efficient photocatalyst.

## Methods

**Photocatalysts synthesis**. PCN was synthesized by the polymerization reaction of cynauric acid-melamine super-molecular assembly under high temperature[36]. In a typical synthesis, 5 g cynauric acid and 5 g melamine was mixed with 180 mL water and 20 mL isopropyl alcohol, and magnetically stirred for 24 h. The white precursor was then collected by centrifugation and dried in vacuum overnight. Dry precursor was calcined at 600 °C for 3 h with temperature ramping rate of 5 °C min$^{-1}$ in muffle furnace with nitrogen flow of 15 L min$^{-1}$. The as prepared yellow sample was washed with water, ethanol, and dried in vacuum oven over night, and then pulverized by ball milling.

Grafting of the sodium cyanaminate moiety on the polymeric carbon nitride framework was realized by molten salt treatment. In a typical synthesis, 1 g PCN and 2 g NaSCN was mixed with a small amount of water for improving the contact of the salt and PCN. The mixture was then dried in vacuum overnight before subject to 400 °C heating in a tube furnace with nitrogen flow. PCN-NaCA-1, PCN-NaCA-2, and PCN-NaCA-3 was, respectively, prepared by mixing 0.5, 1, and 2 g of NaSCN with 1 g PCN under the otherwise same reaction condition.

**Batch photocatalytic reaction**. The photocatalytic hydrogen peroxide production reactions were conducted in jacketed glass reactor filled with 50 mL glycerol aqueous solution (3.5 wt%) and catalyst. The pH of the mixture was adjusted by KOH or HClO$_4$ to the expected value. The photoreactor was irradiated by solar simulator with energy density of 100 mW cm$^{-2}$ on the outer-surface of the jacketed reactor. The reaction mixture was sampled at specific time intervals.

**Continuous serial microbatch reaction**. Two peristaltic pumps were used to feed the photocatalyst suspension and gaseous oxygen. A T-shaped mixing device was used to connect the oxygen feeding tubing and reaction mixture feeding tubing. Continuous gas–solid–liquid triphasic mixture segments are produced and pushed into a coil reactor made of PTFE tubing (Φ1.6 × 3.2 mm) with volume of 92 mL. Gaseous oxygen and photocatalyst suspension were fed at the same flow rate of

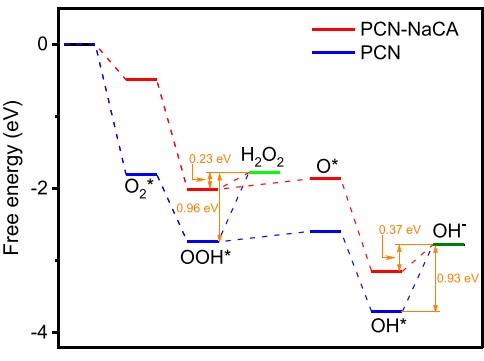

**Fig. 8 Free energy diagrams of ORR (oxygen reduction reaction) steps on PCN-NaCA (red lines) and PCN (blue lines).** PCN polymeric carbon nitride, PCN-NaCA-2 polymeric carbon nitride with sodium cyanaminate moiety.

1.3 mL min$^{-1}$ for retention time of 36 min in the flow reactor; and setting the flow rate of 0.65 mL min$^{-1}$ affords a retention time of 72 min. A column LED array (Household white light LED column) is placed coaxially inside the coil reactor, and the light intensity at the surface of the PTFE tubing was 27 mW cm$^{-2}$.

Reaction mixture preparation: photocatalyst was dispersed in 1000 mL glycerol aqueous solution with concentration of 3.5 wt% by ultrasonication for 30 min in dark. The reaction mixture was magnetically stirred during pumping.

**Ultrafast spectroscopy measurement**. The fs-TAS data was collected on a commercial transient absorption spectrometer (HELIOS, Ultrafast systems). The laser was generated by the 1 kHz Astrella (Coherent Corp.) Ti:sapphire regenerative amplifier (800 nm) and was split to generate pump and probe pulses. The pump pulse passed a TOPAS-Prime (Light Conversion, Ltd.) optical parametric amplifier to generate the 365 nm excitation pulse. The energy density of the excitation pulse was tuned by a natural density filter wheel. The probe light was generated by focusing the 800 nm light through a sapphire crystal. The time resolution of this setup was 120 fs.

Sample preparation: 20 mg sample was dispersed in 20 mL glycerol aqueous solution with concentration of 3.5 wt%. The suspension was ultrasonicated for 12 h, and then centrifuged at a relative centrifugal force (RCF) of 2654×$g$ for 10 min to remove the large carbon nitride aggregates. The homogeneous suspension was subjected to fs-TAS measurement. The aerobic experiments were conducted with oxygen-saturated suspension, and the anaerobic experiments was conducted with high-vacuum treated suspension.

**Electrochemical oxygen reduction reaction (ORR) performance measurement**. The electrochemical ORR performance is evaluated on a CHI 760E workstation equipped with PINE rotating ring disk electrode (RRDE) unit. The electrochemical system setup is assembled based on three-electrode configuration consisting of catalyst-coated RRDE working electrode, Ag/AgCl reference electrode, Pt/C counter electrode, and O$_2$-saturated 0.1 M KOH electrolyte. The working electrode is fabricated by a traditional "drop-casting" method. Specifically, the ink was prepared by mixing 5 mg of catalyst, 750 μL of isopropyl alcohol, 250 μL of water, and 10 μL of Nafion solution (5%) under ultrasonication. Next, 10 μL ink was dropped onto the surface of electrode and dried in air under room temperature. The loading of the catalyst on RRDE electrode is 0.202 mg cm$^{-2}$.

The RRDE has an electrode area of 0.2475 cm$^2$ for disk electrode, 0.1866 cm$^2$ for ring electrode, and a collection efficiency of 37%. The cathodic current density come from disk current normalized by electrode area (0.2475 cm$^2$), and the anodic current density come from ring current normalized by ring electrode area (0.1866 cm$^2$) and collection efficiency. H$_2$O$_2$ production selectivity ($S_{H_2O_2}$) is calculated according to Eq. (1).

$$S_{H_2O_2} = 2 \times \frac{\frac{I_r}{N}}{I_d + \frac{I_r}{N}} \times 100\%, \tag{1}$$

where $N$ is the collection efficiency, $I_r/N$ is the normalized ring electrode current, $I_d$ is disk electrode current.

## Data availability
Source data are provided with this paper. All data are also available from the corresponding authors on request Source data are provided with this paper.

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

## Acknowledgements

The financial supports from National Natural Science Foundation of China (NOs. 21976041, 51538013, 51838005) and Leading Researcher Program through the National Research Foundation of Korea (NRF) (NRF-2020R1A3B2079953) are acknowledged. We are grateful to Prof. Kaifeng Wu from Dalian Institute of Chemical Physics, Chinese Academy of Sciences and Prof. Guigang Zhang from Fuzhou University for fruitful discussions. The authors appreciate the XAS measurements from Singapore Synchrotron Light Source (SSLS) SUV (Soft X-Ray-Ultraviolet) beamline and Shanghai Synchrotron Radiation Facility (SSRF) BL08U1A beamline.

## Author contributions

Y.Z. conceived the project and designed the experiments; Y.Z., P.Z., Z.Y., L.L., J.G., S.C., T.X., C.D., S.X., and B.X., conducted the experiments and data analysis; Y.Z., P.Z. S.C., and W.C. wrote the original draft; Y.Z., C.H., and W.C. jointly supervise the project. All authors commented on the final manuscript.

## Competing interests

The authors declare no competing interests.
