## [Peer Review File · Nature Communications]

Reviewers' comments:

Reviewer #1 (Remarks to the Author):

This paper deals with the overall Solar-driven photocatalytic transformation process of hydrogen peroxide. Before the paper can be published 10 points should be clarified:

1. The solar conversion efficiency bottleneck should be briefly explained in the introduction. You could include which is the currently highest efficiency achieved (annotated in the supplementary information document). I see that the explanation of why is it low is in the results reaction, when it should be in the introduction section. Is that the only important one? What are others bottleneck for solar panels / fuels study? Cost? Affordable for middle class people?
2. I miss a materials/methods separate section from the results section. I know that the synthesis information is in the supplementary information document, but it would be nice to have a small section of materials/methods in the paper as well.
3. How the content of Na is calculated from the XPS showed in the supplementary information? In the document is indicated "the content of sodium in 168 PCN-NaCA-2 is 3.99 (atom.) % as determined by XPS" I do not see how is this achieved.
4. Continuing with XPS, in the N1s spectrum, I really believe that what is fitted as just one peak in the 400.1-eV region should be definitely fitted with more than 1, as the shape of a peak doesn't properly fit the spectrum (top spectrum of the d graph). Can this N1S spectra be revised?
5. Porosity measurements are highly recommended to increase value to the SEM and TEM images observed in the supplementary information document.
6. The are no differences in the SEM/TEM images when comparing the different photocatalyst compounds prepared (PCN-NaCA-1, PCN-NaCA-2, and PCN-NaCA-3)? A morphology comparison would be useful. Do you assume they have the same surface active area, same morphology the same porosity?
7. EDS showing the uniform distribution of sodium in the matrix doesn't appear in the Supporting Information, should be included to corroborate the statement
8. In the Femtosecond-TAS spectroscopy, when you perform the measures in in pure O₂ atmosphere, how do you know that you have surface adsorbed dioxygen molecules? Did you chemically characterize the surface of the photocatalyst in this atmosphere? You do not know how the surface looks like if you do not characterize it first. Did you observe these molecules? Are you sure that it's just O₂ adsorbed molecules and you are not chemically/physically modifying the surface in this atmosphere? It should be useful to chemically characterize the surface of the photocatalyst under these conditions before performing the measures.
9. Can the author clarify which useful information or conclusion is extracted from the transient photovoltage (TPV) characterization? It seems that it was included in the article because it was done more than because its relevant to the project
10. There is no information about the temperature programmed oxygen desorption (O₂-TPD) technique, how it was performed. Why the area of the desorption peak indicates the amount of dioxygen adsorbed per unit catalyst mass? Why those peaks are not related to other changes happening in the surface during the grading of temperature of the experiment?

In summary, the authors perform a huge number of techniques, some of them more useful and with more clear conclusion than others, where they didn't even clarified why they were performed. The article is really complete. However, there are some points that should be revised, like the XPS fitting parameters, the STEM-EDS images, and the chemical characterization of the surface in O₂ rich environments before the Femtosecond-TAS spectroscopy is performed.

Reviewer #2 (Remarks to the Author):

The authors present a study focused on using modified carbon nitride materials as a platform for supporting photon-driven hydrogen peroxide production via oxygen reduction. While the justification for this study is reasonable and the findings are somewhat novel, there are a few technical areas wherein I'm not convinced that some of the conclusions drawn by the authors are fully justified by the data presented in the manuscript. As such, I would suggest that a revised version of this manuscript be reconsidered after the authors have considered the technical points outlined below.

The authors claim that their sodium cyanamate treatment creates p-type and n-type regions. I am somewhat skeptical that this cyanamate treatment will indeed engender sufficiently high doping densities to achieve true n and p regions in the traditional sense. I find it instead more likely that the resulting material that the authors have obtained comprises a larger distributions of sheet sizes or particle sizes that may lead to a distribution in the energy landscape that causes additional charge localization following photoexcitation.

Why are the Delta OD vs. Fluence values in Figure 6 plotted on a log-log scale? This seems like a very strange way to plot data that the authors are trying to use to claim that they observe a linear TA response with excitation fluence, particularly, since the data barely constitute a single decade of values. Moreover, three data points are generally insufficient to support the assertion that there are no two-photon effects contributing to the observed optical signal when using such short laser pulses with high peak powers.

The authors say that the surface photovoltage signal is proportional to the amount of trapped charge. However, this doesn't seem correct. It seems like the photovoltage signal will likely either be quadratic or logarithmic with charge density depending on what the energy profile of the density of tail states looks like. In any event, I believe that this assertion warrants further discussion.

Reviewer #3 (Remarks to the Author):

In this paper, the authors describe the utilization of modified carbon nitride as photocatalyst for oxygen reduction to hydrogen peroxide. The results are encouraging and the authors seriously attempt to reveal the reaction mechanism. The weak points are the materials synthesis and final structure, stability studies (including reasons for possible instability of the photocatalyst), lack of other and more sustainable hole scavengers, and other points as state below. Overall, I think that the work has the potential to be published in Nature Comm after rigorous revision.

1. The part of the synthesis should be better elucidated. Usually, molten salts are inorganic salts that are used as reaction media and for templating. Here, it is a different case. The proposed structure is more of an assumption here. The authors may consider learning it at different condensation temperatures and to include TGA coupled with GC-MS to understand the growth.
2. One weak point of this study is the use of glycerol as a hole scavenger. Glycerol is not such a cheap reagent. I think that other oxidation reaction (water splitting, pollutant degradation, organic molecule oxidation to valuable reagents, etc.) should be studied as an alternative.
3. How much Na⁺ do they have in the final materials? Is it within the CN framework or more on the surface of the material?
4. A comparison to PCN after heat treatment at 400 C should be given as well. In addition, how can the authors exclude the creation of N vacancies after post-heating to 400 C (as shown many times before)? The latter may be the reason for the absorption shift and enhanced catalytic activity.
5. Long-term stability measurements are missing, including post-characterization and recycling of the catalyst.

6. The authors state that the absorption is better in the range of 450 – 500 nm. However, according to the AQE, there is no activity in this region. What are the reasons for that? Usually, this enhanced absorption can be attributed to defects states below the conduction band.
7. How can they avoid the oxidation of Glycerol by hydroxyl radicals? It may inhibit that reaction after a short time as the peroxide will be consumed.
8. The mechanism of glycerol degradation should be studied.
9. A summary of the outcome of the TAS and SPV will be useful for the reader (also as an illustration).
10. The title is slightly misleading – in this paper the authors don't really identify the factor for...

Reviewer #4 (Remarks to the Author):

This study reported a molten salt method that can introduce sodium cyanamate (NaCA) moiety to polymeric carbon nitride (PCN) frameworks. The authors then investigated the effect and mechanisms of NaCA of PCN on the photochemical synthesis of H₂O₂ in glycerol solution under simulated solar light. I would recommend it for publication in Nature Communications after the following critical issues are solved.

1. Fig.S1e the NMR of PCN-NaAC-2 did not show any obvious differences compared with that of the PCN sample. Can the author enlarge the specific part to show the peak at 171 ppm?
2. The author used 0.38M glycerol aqueous solution for H₂O₂ production and claimed the glycerol worked as electron and proton donors. I recommend the analysis of the final oxidation product of glycerol to get solid evidence for this conclusion. For instance, how much of the raw glycerol is oxidized to glyceraldehyde after the photochemical reaction? Can the concentration of glycerol be lower considering that 0.38M is quite high?
3. Following the previous question, the use of glycerol as sacrifice agent would increase the cost and bring impurities to the final H₂O₂ production. Glycerol itself is a useful chemical. Do the authors think that the produced H₂O₂ will be more valuable? Besides, how to purify the H₂O₂ from the final mixture for further usage?
4. The H₂O₂ decomposition also happens in photocatalysis (e.g. oxidized by holes) and in alkaline solution. The authors are suggested to conducted experiments to investigate the decomposition rates of PCN and PCN-NaAC-2.
5. Line 102. What is the flow rate in the continuous serial micro-batch reactor? More details should be provided in the SI, such as catalyst suspension flow rate, O₂ flow rate and the light wavelength and intensity of column LED array?
6. Line 249-260. On the 2e⁻ ORR for H₂O₂ production, it can further be classified to one-step direct 2e⁻ ORR and two-step indirect 2e⁻ ORR (forming superoxide radical first) by PCN. According to previous studies (e.g. Chem. Commun., 2019, 55, 13279-13282; ACS Catal., 2020, 10, 3697-3706), the indirect 2e⁻ ORR process can also promote the generation of H₂O₂ production. The authors need to further conduct some experiments(For example EPR spectroscopy) to see whether there were superoxide anions generated in photocatalysis. This analysis can further understand the photocatalysis reaction mechanism.
7. Line 269. Can the author explain why the theoretical simulation of 2e⁻ ORR steps were performed in alkaline solution, but not acid or neutral solution?

Reviewer #5 (Remarks to the Author):

This is a detailed investigation (mostly experimental, with some calculations) on the photocatalytic evolution of H₂O₂ using polymeric carbon nitride (PCN) based photocatalysts that have been treated with NaSCN in order to add sodium cyanamate moieties. The paper shows an enhancement of H₂O₂ evolution over non-treated PCN materials. The enhancement is quantified by a table in the SI, showing an approximately sixfold increase in H₂O₂ production rate over the past attempts show, a list which is restricted to different varieties of PCN.

I have a variety of questions and criticisms of the paper. Some are fundamental and some are technical.

1) I do not see a clearly quantified reason why this study reflects a high-profile advance (Nature Communications) rather than a contribution to the chemical literature that is, while valuable, essentially a technical report. It is not clear that the reported sixfold increase will make a difference between what is an interesting technical observation and a true advance that would (as indicated in the introduction) sway industry away from its conventional processes. How does the process compare to other processes in the literature? The substances used/synthesized here are not new, so there is no new understanding of the fundamental processes and/or components of the system either.

2) The mechanistic interpretation of the results, while approached for multiple angles, is, ultimately, not stringent. I recognize that this is a difficulty that anyone working on PCN materials must face. The atomic structure of the actual materials in question, especially the nature of active sites, is simply not understood in detail - nor is it possible to determine this atomic structure with certainty by any existing experimental or computational techniques, to the best of my understanding. Any model must therefore remain qualitative and speculative, except for the overall observations that can be extracted directly from experiment or from theory.

But even given these difficulties, some questions remain inherently open based on the present paper.

2a) The paper assumes that O₂ is being converted to H₂O₂ - but no evidence is given that it is indeed O₂, and not something else, that is being converted. What is the source of this O₂? What is its partial pressure / concentration? There is glycerol in this system. Why is it not glycerol that reacts and, in the process, releases H₂O₂? There are other interesting ingredients in the system (KOH, HClO₄, perhaps others) and the role of the oxygen-containing anions is not discussed.

So, evidence that it is indeed O₂ that is being converted, and not something else, should be provided.

But in any case, I missed what is the O₂ source and how it is controlled. In a quantitative chemical study, it would be essential to control, characterize and report the reagents appearing on either side of the reaction.

2b) Structural assumptions made in the discussion of the material. The paper is careful in its text descriptions of PCN derived materials. In particular, it is good to see that a variety of melon, rather than a hypothetical H-free C₃N₄ material, forms the foundation of at least the computational understanding.

Nevertheless, Figure 1a shows a depiction of a hypothetical structure of PCN-NaCA-n that is, in my view, unsubstantiated in this paper and has been subject to an extensive discussion in the literature. The crux is that the C₃N₄ like structure shown here likely cannot be made and, to my knowledge, there is no firm evidence in the literature that it ever has been made.

Among the references cited (indirectly) on this point: Refs 37, 38, and 47, which are used as

evidence, are early and predate this debate. In these papers, the existence of a fully C₃N₄ like condensate was essentially a plausible assumption, but later debunked. Ref. 35 is a review and Ref 36 also presents no structural evidence. Ref. 35 states specifically: "Unfortunately, the crystal structure of heptazine-based CN is still not very clear."

Furthermore, there is at least one reference (Chemistry of Materials 29 (10), 4445-4453) that shows from simple thermodynamic considerations that the C₃N₄ like structure shown Figure 2a (right) cannot arise under plausible thermodynamic conditions. In short, I think the unsubstantiated structural hypothesis of heptazine based C₃N₄ condensation (Fig 2a right) should be avoided and a more open nanostructure (like melon/PCN, Fig 2a middle or some similar intermediate) should be the foundation of the discussion and figures.

2c) Similar for "the improved polymerization degree increases layer buckling, ..." in the discussion ... there is no evidence for this in the paper.

2d) I agree with much of the qualitative discussion of the experimental results (trapped charges, charge migration). One thing that remains unclear to me is the SPV spectrum in Fig 7b. The abrupt onset at 0.8 μ -s is not clearly explained and the explanation in terms of charge diffusion from bulk to surface is not clear to me at all. The sharp onset does not look like diffusion, which is a gradual process. It is difficult to understand why there is some time constant here with no signal at all.

2e) I believe that the computational data shown are inconclusive and not well substantiated. I should prequel this by saying that I understand well that the models itself need to remain conceptual since no experimental structural evidence regarding the actual reaction site or atomic geometry exists at all, nor is it clear which statistical ensemble of reaction sites should be considered. So the conceptual models themselves are certainly necessary and the existence of the PCN sites shown is not implausible.

However, even then, there are several unsubstantiated assumptions and technical omissions in the theory, which raise doubts:

- How plausible is the presence of Na⁺ in the site shown? This is an equilibrium with H₂O, or so I understand, and Na⁺ is rather soluble. What is the expected concentration of Na⁺ in such a site, compared to Na⁺ in solution? There are multiple other ionic species around. A thermodynamic analysis of the probability of Na⁺ occupying the sites in question should be provided. (The presence of Na⁺ is a critical prerequisite for the energetics claimed and so this assumption should be substantiated.)
- What is the spin state of the adsorbed O₂? This would be important.
- What is the spin state of any other intermediates considered?
- How were the O₂ adsorption sites determined? There are many possible adsorption sites - was a search for other possible adsorption sites performed?
- What are the charges found on individual species in these simulations? are they physical? If these overall simulation cells are electrically neutral, the presence of different ionic moieties still implies a significant role of charge transfer. However, the PBE functional used here (or any GGA) is known to describe charge transfer unphysically.
- In particular, the drastic increase of the O₂ adsorption energy in the presence of Na⁺ / NCN- remains unexplained from a fundamental point of view. Given that the O₂ molecule bonds so strongly, some significant charge transfer is likely. But this charge transfer could be entirely an artifact of the simulation - again, charge transfer is simply not described correctly by GGAs. A much more

exhaustive analysis of the simulations would be necessary to clarify these issues.

- Is 2 nm of "vacuum" really enough to electrostatically decouple different sites with strong dipolar characteristics from one another? 2 nm is not much at all. There should be some electrostatic interaction between the supercells considered.

- There is no "van der Waals interaction" in the PBE functional (or in any GGA). This is very well known. Given that the bonding here has van der Waals character at least in parts, their absence is a technical error that could invalidate the simulations altogether.

- Figure 8 shows "free energies". And indeed, given that the energies of several reagents are partial pressure / concentration dependent, free energies should be used.

However, the authors say nothing regarding partial pressure of O₂ during the reaction, either in experiment or in the theory part. The correct way to couple the free energies of reference gas phases into such simulations is well known, e.g., Physical Review B 65 (3), 035406. Did the authors do this? If not, the numbers should be corrected and the appropriate analysis should be provided.

- Without availability of the geometries used in the computations (all computational steps) the simulations will not be reproducible. Since numerous public repositories are now available to deposit such information, all pertinent input files should be made available.

Point-to-Point Response to Reviewers' Comments

(Manuscript NO.: NCOMMS-20-36228)

Reviewer #1 (Remarks to the Author):

Comments:

"This paper deals with the overall Solar-driven photocatalytic transformation process of hydrogen peroxide. Before the paper can be published 10 points should be clarified."

Responses:

We appreciate the reviewer for these constructive comments. We have made revisions on the manuscript, and addressed all the concerns from the reviewer.

Comments:

"1. The solar conversion efficiency bottleneck should be briefly explained in the introduction. You could include which is the currently highest efficiency achieved (annotated in the Supplementary Information document). I see that the explanation of why is it low is in the results reaction, when it should be in the introduction section. Is that the only important one? What are others bottleneck for solar panels / fuels study? Cost? Affordable for middle class people?"

Responses:

- (1) The gap between solar energy potentials and the practical application of it is the cost-effectiveness of the solar conversion systems. The cost-effectiveness relies on the low cost and high efficiency of the materials on which the photon energy is converted/stored. We have revised manuscript and discussed this aspect in the introduction part.
- (2) In solar cell studies, there are standard equipment/methodologies for the evaluation of solar conversion efficiency, and the efficiency data are available and comparable among most of the reports. We strongly recommend adding efficiency data, e.g., apparent quantum yield, for evaluating the photocatalytic performance of a material in a chemical conversion reaction. It is, however,

unavailable in most of the literature reports. We thus just listed the efficiency data available in Table S1.

Comments:

“2. I miss a materials/methods separate section from the results section. I know that the synthesis information is in the Supplementary Information document, but it would be nice to have a small section of materials/methods in the paper as well.”

Responses:

METHODS section is added in the revised manuscript, in which selected materials/methods information is described in detail.

Revisions:

Revisions in METHODS section in the manuscript on page 16-17:

METHODS

Photocatalysts synthesis: ...

Batch photocatalytic reaction: ...

Continuous serial micro-batch reaction: ...

Ultrafast spectroscopy measurement: ...

Electrochemical oxygen reduction reaction (ORR) performance measurement: ...

Comments:

“3. How the content of Na is calculated from the XPS showed in the Supplementary Information? In the document is indicated “the content of sodium in 168 PCN-NaCA-2 is 3.99 (atom.) % as determined by XPS” I do not see how is this achieved.”

Responses:

(1) The atomic percentages of the elements were calculated based on the integration of the C 1s, N 1s and Na 1s peaks by software; and it only reflects the composition of the surface.

(2) In the revised manuscript, we employed ICP-OES to analyze the overall sodium content in the sample; and the weight percentage of sodium in PCN-NaCA-2 is 6.9 %.

Revisions:

Revisions in the manuscript on page 3:

The sodium content of PCN-NaCA-2 was determined to be 6.9 wt.% by inductively coupled plasma atomic emission spectroscopy (ICP OES).

Comments:

"4. Continuing with XPS, in the N1s spectrum, I really believe that what is fitted as just one peak in the 400.1-eV region should be definitely fitted with more than 1, as the shape of a peak doesn't properly fit the spectrum (top spectrum of the d graph). Can this N1S spectra be revised?"

Responses:

Following the reviewer's suggestions, N 1s peaks in the spectra were deconvoluted into three peaks (398.6, 400.0, and 401.2 eV), and the fitting curve matches much better with the original N 1s signals (Figures S1c and S1d). The interpretation of the XPS spectra is revised as well in the supporting information.

Revisions:

Revised Figure S1b in Supplementary Information on page S8:

Comments:

“5. Porosity measurements are highly recommended to increase value to the SEM and TEM images observed in the Supplementary Information document.”

Responses:

The samples were characterized with nitrogen physisorption. As shown in Figure S4, PCN has pores with sizes of 2 – 6 nm, and PCN-NaCA-1 presents sharper size distribution curve centered at 3.5 nm. and PCN-NaCA-2 has similar pore size distribution to PCN, but much lower BET surface area, e.g., 83.2 m²/g for PCN versus 11.9 m²/g for PCN-NaCA-2. PCN-NaCA-3 shows wide pore size distribution and BET surface area of 18.1 m²/g. The surface area and pore structure are not decisive factors for the photocatalytic H₂O₂ production performance, as PCN-NaCA-2 shows the lowest BET surface area, but highest photocatalytic activity among all the photocatalysts.

Revisions:

Figure S4 in Supplementary Information on page S12:

Figure S4. BJH Pore size distribution. (a) PCN, (b) PCN-NaCA-1, (c) PCN-NaCA-2, (d) PCN-NaCA-3.

Comments:

“6. The are no differences in the SEM/TEM images when comparing the different photocatalyst compounds prepared (PCN-NaCA-1, PCN-NaCA-2, and PCN-NaCA-3)? A morphology comparison would be useful. Do you assume they have the same surface active area, same morphology the same porosity?”

Responses:

SEM/TEM images (Figure S3), BET surface area (Table S1), and pore size distribution (Figure S4) information have been added in the revised supporting information. From the SEM/TEM characterizations, the morphology differences in micrometer scale is not obvious. By HRTEM, PCN-NaCA-2 is the only sample that could be observed with clear layer stacking structures in nanometer scales.

Revisions:

(1) Figure S3 and notes in Supplementary Information on page S11:

Figure S3. SEM (a, d) and HRTEM (b, c, e, and f) images of PCN-NaCA-1 (a, b, c) PCN-NaCA-3 (d, e, f).

Notes: SEM images shows that these samples have similar morphology; and due to the poor crystallinity of these samples, layer-stacking structures are not observed by HRTEM.

(2) Table S1 in Supplementary Information on page S12:

Table S1. BET surface area of the samples.

Sample	BET surface area (m ² /g)
PCN	83.2

PCN-NaCA-1	95.6
PCN-NaCA-2	11.9
PCN-NaCA-3	18.1

Comments:

“7. EDS showing the uniform distribution of sodium in the matrix doesn’t appear in the Supporting Information, should be included to corroborate the statement”

Responses:

It is in Figure S1, panels g and h show the EDS mapping images of sodium and nitrogen elements, respectively, in sample PCN-NaCA-2.

Comments:

“8. In the Femtosecond-TAS spectroscopy, when you perform the measures in in pure O2 atmosphere, how do you know that you have surface adsorbed dioxygen molecules? Did you chemically characterize the surface of the photocatalyst in this atmosphere? You do not know how the surface looks like if you do not characterize it first. Did you observe these molecules? Are you sure that it’s just O2 adsorbed molecules and you are not chemically/physically modifying the surface in this atmosphere? It should be useful to chemically characterize the surface of the photocatalyst under these conditions before performing the measures.”

Responses:

(1) There are technologies for observing the molecules on a surface with atomic-level roughness, such as scanning tunneling microscope (STM). While on the surface of nanoparticles, such as the case here, observing a surface-adsorbed dioxygen molecule is rather challenging, to the best of our knowledge. We are unable to clearly describe the details on the surface of a nanoparticle in atomic-level at the current stage.

(2) In this work, the samples were prepared, stored, and handled in air, i.e., chemical/physical

modification of the surface by oxygen, if any, has already happened before the spectroscopy measurement. We discovered the significant differences in transient absorption spectra under vacuum and oxygen atmosphere. These spectroscopic observations experimentally verify the impact of dioxygen on the excitation and charges trapping process of the photocatalyst, which is recently described in an theoretical simulation investigation (*Nat. Commun.* **12**, 320 (2021)).

Comments:

“9. Can the author clarify which useful information or conclusion is extracted from the transient photovoltage (TPV) characterization? It seems that it was included in the article because it was done more than because its relevant to the project.”

Responses:

- (1) The amount and the life-time of charge carriers are critical to an efficient photocatalytic oxygen reduction reaction. We thus employed TPV for monitoring the charges recombination behavior dynamically. It is found that the photovoltage signal can last for 2.5 ms, which is much longer than that of PCN (0.6 ms). Accordingly, the depletion of charge carriers by recombination is significantly attenuated by construction of the sodium cyanamate moiety on polymeric carbon nitride framework. This result is conducive for understanding the rationale behind the superior performance of PCN-NaCA-2 in photocatalytic H₂O₂ production.
- (2) In the revised the manuscript, we have added more discussions on the TPV results as well as the implications to the photocatalytic performance.

Revisions:

- (1) Revisions in the manuscript on page 13:

The dynamics behavior of the charge carriers is further investigated by transient photovoltage (TPV) characterization. For both PCN and PCN-NaCA-2, there are two photovoltage peaks (peak a and peak b, Figure 7b), which are, respectively, attributed to drift and diffusion of the photo-induced charges.^{83,84} PCN-NaCA-2 presents a remarkably strong and negative peak b lasting for 2.5 ms before decaying to zero, speaking for the diffusion of large number of photo-induced electrons to the surface after photons excitation event. The photovoltage (PV) characterizations demonstrate, obviously, that the depletion of the charge carriers by recombination is significantly attenuated by the construction of the sodium cyanamate moiety on polymeric carbon nitride framework.

References:

83. Mora-Seró, I., Dittrich, T., Garcia-Belmonte, G. & Bisquert, J. Determination of spatial charge separation of diffusing electrons by transient photovoltage measurements. *J. Appl. Phys.* **100**, 103705 (2006).
84. Kronik, L. & Shapira, Y. Surface photovoltage phenomena: theory, experiment, and applications. *Surf. Sci. Rep.* **37**, 1-206 (1999).

(2) Revised Figure 7b in the manuscript on Page 13:

Comments:

“10. There is no information about the temperature programmed oxygen desorption (O₂-TPD) technique, how it was performed. Why the area of the desorption peak indicates the amount of dioxygen adsorbed per unit catalyst mass? Why those peaks are not related to other changes happening in the surface during the grading of temperature of the experiment?”

Responses:

- (1) We apologize for missing the information on O₂-TPD measurement. In the revised supporting information, there is detailed descriptions on O₂-TPD measurement.
- (2) Inspired by the reviewer's comments, we conducted TGA-IR-MS analysis on the sample under oxygen atmosphere to check the possibility of other changes on the surface. As shown in Figure S31, there is a small weight loss of 2.1 % in the TG curve. The infrared spectra observe the release of H₂O during heating, and the mass spectra also confirms that water is the only species evolved during heating. The evolved H₂O comes from the surface adsorption, since the sample was stored in air and did not pre-treated before measurement.

- (3) The samples were prepared by polymerization reaction under high temperatures (e.g., 400 °C or 600°C, please refer to METHODS in the revised manuscript). In the O₂-TPD measurement, the sample was pretreated at 300 °C in helium flow to remove the surface adsorbates. After the adsorption equilibrium with a O₂(2 %)/He pulse, the physically adsorbed oxygen was removed by He flow; the TPD measurement then started with the temperature program, and desorbed oxygen was monitored by thermal conductive detector (TCD).

Revisions:

- (1) Revisions in Supplementary Information on page S4:

Oxygen temperature programmed desorption (O₂-TPD) was measured on Micromeritics AutoChem II 2920. The sample was pretreated in He flow under 300 °C for 1 hour; and a pulse of 2 % O₂ in He was used for absorption of the oxygen molecules on the sample, followed by 1 h He flow with flow rate of 50 mL/min at 50 °C for removing the physically adsorbed oxygen molecules. O₂-TPD was measured in the He flow with rate of 50 mL/min; the initial temperature was 50 °C, and the ramp rate was 15 °C/min. Desorbed oxygen was monitored by thermal conductivity detector (TCD).

- (2) Figure S31 in Supplementary Information on page 30:

Figure S31. TG-IR-MS characterization of PCN-NaCA-2 in aerobic condition. (a) TG curve; (b) IR spectra of the evolved gases with temperature ramping; (c) MS signals of selected ions with rising temperature, CO₂ (m/z = 44) and H₂O (m/z = 17 and 18).

Notes: There is a small weight loss of 2.1 % in the TG curve. The infrared spectra observe the release of H₂O during heating, and the mass spectra also confirms that water is the only species evolved during heating. The evolved H₂O comes from the surface adsorption, since the sample was stored in air and did not pre-treated before measurement. These results indicate that pretreatment in helium flow at 300 °C is enough for removing the surface adsorbed molecules in O₂-TPD.

Comments:

“In summary, the authors perform a huge number of techniques, some of them more useful and with more clear conclusion than others, where they didn’t even clarified why they were performed. The article is really complete. However, there are some points that should be revised, like the XPS fitting parameters, the STEM-EDS images, and the chemical characterization of the surface in O₂ rich environments before the Femtosecond-TAS spectroscopy is performed.”

Responses:

- (1) For an in-depth understanding of critical factors governing the solar conversion efficiency, a series of techniques were employed for a step-by-step analysis of the complicated processes involved in the photocatalytic H₂O₂ production, e.g., photons absorption, charge separation and diffusion, and surface reactions.
- (2) In the revised manuscript, we have added more necessary explanations to clarify the reasons and conclusions.

Revisions:

- (1) Revisions in manuscript on page 10:

As the non-emissive trapped electrons have the potential for participating chemical conversion reaction on the surface. We then focus on the status of the non-emissive trapped electrons and their interaction with the surface adsorbed dioxygen on PCN and PCN-NaCA-2.

- (2) Revisions in manuscript on page 12:

For initiating oxygen reduction reactions, the photo-induced charge carriers must diffuse to the surface active-sites. The amount of the photo-induced charges and the charges diffusion properties are of great importance, and thus be investigated by surface photovoltage (SPV) spectroscopy.

- (3) Revisions in manuscript on page 13:

The photovoltage (PV) characterizations demonstrate, obviously, that the depletion of the charge carriers by recombination is significantly attenuated by the construction of the sodium cyanamate moiety on polymeric carbon nitride framework.

- (4) Revisions in manuscript on page 14:

It is difficult to further reveal the reaction mechanism by monitoring the transient intermediates on the surface via experimental methodologies. However, computational quantum chemistry offers a platform that makes the prediction of the surface-structure dependent reaction mechanism feasible.⁸⁹ Density functional theory (DFT) calculation of the ORR reaction mechanism (both 2e⁻ and 4e⁻ ORR pathways) is therefore performed for further understanding the rationale behind the superior performance of PCN-NaCA-2 for H₂O₂ production.

Reference:

89. Nørskov, J. K., Bligaard, T., Rossmeisl, J. & Christensen, C. H. Towards the computational design of solid catalysts. *Nat. Chem* **1**, 37-46 (2009).

Reviewer #2 (Remarks to the Author):

Comments:

“The authors present a study focused on using modified carbon nitride materials as a platform for supporting photon-driven hydrogen peroxide production via oxygen reduction. While the justification for this study is reasonable and the findings are somewhat novel, there are a few technical areas wherein I’m not convinced that some of the conclusions drawn by the authors are fully justified by the data presented in the manuscript. As such, I would suggest that a revised version of this manuscript be reconsidered after the authors have considered the technical points outlined below.”

Responses:

We appreciate the reviewer for these valuable comments. We have carefully considered the comments and made intensive revisions accordingly, as shown in the revised manuscript and Supplementary Information.

Comments:

The authors claim that their sodium cyanamate treatment creates p-type and n-type regions. I am somewhat skeptical that this cyanamate treatment will indeed engender sufficiently high doping densities to achieve true n and p regions in the traditional sense. I find it instead more likely that the resulting material that the authors have obtained comprises a larger distributions of sheet sizes are particle sizes that may lead to a distribution in the energy landscape that causes additional charge localization following photoexcitation.”

Responses:

(1) We appreciate the reviewer’s very constructive comments. We fully understand the reviewer’s skepticism on the description of conductivity-type in the manuscript; and we are very interested in the proposed mechanism by the reviewer on the impact of sheet/particle sizes on the distribution of energy landscape. We, indeed, would like to explain the data from the reviewer’s perspective, however, the theoretical or experimental basis is not enough for supporting such a novel proposal so far.

- (2) In this work, the sodium content in PCN-NaCA-2 was determined to be 6.9 wt.%, stating that the amount of sodium cyanamate moiety in the polymeric carbon nitride framework is not negligible. We measured the M-S curve in dark condition; and most importantly, the correlation between the slope of M-S curve and the conductivity-type is well-established. We therefore explain the experimental data accordingly.
- (3) We will keep the reviewer's mechanism in mind, and further consider it in the future work based on available experimental/theoretical data. Meanwhile, we introduced this idea partially in the revised manuscript.

Revisions:

(1) Revisions in the Abstract:

The structural features of cyanamino group and pyridinic nitrogen-coordinated sodium in PCN-NaCA promote photon absorption, alter the energy landscape of the framework and improve charge separation efficiency, enhance surface adsorption of dioxygen, and creates highly selective $2e^-$ ORR surface-active sites.

(2) Revisions in the manuscript on Page 3:

Introduction of sodium cyanamate moiety by molten-salt treatment creates the electron-withdrawing cyanamino-group and coordinative interaction between sodium and pyridinic nitrogen, and thus alters the distribution of the energy landscape of the carbon nitride framework.

(3) Revisions in the manuscript on Page 13:

The p-type conductivity might be attributed to the strong electron withdrawing property of the cyanamino-group and the coordinative interaction between sodium and pyridinic nitrogen in the framework. These structural features could lead to a unique energy landscape that may promote the charge separation under illumination.

(4) Revisions in the manuscript on Page 15:

...altering the energy landscape of the framework and leading to retarded radiative charge recombination and improved electrons accumulation in the surface region,

Comments :

"Why are the Delta OD vs. Fluence values in Figure 6 plotted on a log-log scale? This seems like a very

strange way to plot data that the authors are trying to use to claim that they observe a linear TA Responses with excitation fluence, particularly, since the data barely constitute a single decade of values. Moreover, three data points are generally insufficient to support the assertion that there are no two-photon effects contributing to the observed optical signal when using such short laser pulses with high peak powers.”

Responses

- (1) Considering the reviewer’s comments, the plots is in linear scale in the revised Figure 6.
- (2) We agree with the reviewer that two-photon excitation could happen under high power density, and more data points are usually necessary. However, in this case, as shown in Figure S27, the data plots significantly deviate from a quadratic scaling as would be expected for two-photon excitation, indicating that two-photon excitation is not relevant to the power densities used in our experiment.

Revisions:

- (1) Revised Figure 6 in the manuscript on page 11:

Figure 6. The fluence dependence of the initial amplitude (black plots) and the decay half-life time (blue plots) of the photo-induced electrons of PCN-NaCA-2 in glycerol aqueous solution in pure oxygen atmosphere and vacuum conditions.

- (2) Figure S27 and notes in Supplementary Information on page S28:

Figure S27. The relevance of the power densities to the theoretical quadratic scaling. (a) with dioxygen; (b) under vacuum.

Notes: the data plots significantly deviate from a quadratic scaling as would be expected for two-photon excitation, indicating that two-photon excitation is not relevant to the power densities used in our experiment.

Comments:

“The authors say that the surface photovoltage signal is proportional to the amount of trapped charge. However, this doesn’t seem correct. It seems like the photovoltage signal will likely either be quadratic or logarithmic with charge density depending on what the energy profile of the density of tail states looks like. In any event, I believe that this assertion warrants further discussion.”

Responses:

- (1) We appreciated this constructive comment, and apologize for our wrong description on the SPV signal intensity.
- (2) The structure of the polymeric carbon nitride is complicated, and the description of the excitation behaviour by mathematical models is so far unavailable. At the current stage, we can only roughly analyse the differences in charge carrier amount and diffusion properties between samples by comparing the intensities of their SPV signals.
- (3) We revise the description on the intensity of SPV signal as shown below.

Revisions:

Revisions in the manuscript on page 12:

The intensity of SPV signal is positively related with the amount of photo-induced charges as well as

spatial charge separation properties; and the sign (positive or negative) of photovoltage is correlated to the direction of the charge diffusion.^{76,77}

References:

76. Mora-Seró, I., Dittrich, T., Garcia-Belmonte, G. & Bisquert, J. Determination of spatial charge separation of diffusing electrons by transient photovoltage measurements. *J. Appl. Phys.* **100**, 103705 (2006).

77. Kronik, L. & Shapira, Y. Surface photovoltage phenomena: theory, experiment, and applications. *Surf. Sci. Rep.* **37**, 1-206 (1999).

Reviewer #3 (Remarks to the Author):

Comments:

“In this paper, the authors describe the utilization of modified carbon nitride as photocatalyst for oxygen reduction to hydrogen peroxide. The results are encouraging and the authors seriously attempt to reveal the reaction mechanism. The weak points are the materials synthesis and final structure, stability studies (including reasons for possible instability of the photocatalyst), lack of other and more sustainable hole scavengers, and other points as state below. Overall, I think that the work has the potential to be published in Nature Comm after rigorous revision.”

Responses:

We acknowledge the reviewer for the constructive comments and the manuscript is improved after an intensive revision based on the reviewer’s suggestions.

Comments:

“1. The part of the synthesis should be better elucidated. Usually, molten salts are inorganic salts that are used as reaction media and for templating. Here, it is a different case. The proposed structure is more of an assumption here. The authors may consider learning it at different condensation temperatures and to include TGA coupled with GC-MS to understand the growth.”

Responses:

- (1) We analyzed the evolved gases during the materials synthesis process on thermal gravimetric analyzer-gas chromatograph-mass spectrometer (TGA-GCMS). The major components of the evolved gases were identified. We thus proposed a conversion mechanism based on the results and the related organic conversion reactions in literatures.
- (2) The results are discussed in the revised manuscript, and Figures S6-S8 are added in the Supplementary Information.

Revisions:

- (1) Revisions in the manuscript on page 3:

By thermal gravimetric analysis-gas chromatography mass spectrometry (TGA-GC-MS), the major components of the evolved gases during synthesis were identified (Figures S5-S7); and the mechanism of the conversion of amino group to cyanamate moiety was proposed accordingly (Figure S8).

- (2) Figures S5 in Supplementary Information on pages 13:

Figure S5. Thermal gravimetric analysis (TGA) curve from the PCN-NaCA-2 synthesis process monitored by thermal gravimetric analyzer-gas chromatograph-mass spectrometer (TGA-GC-MS). The precursor (PCN and NaSCN) for PCN-NaCA-2 was heated under nitrogen flow in an alumina crucible of the thermal gravimetric analyzer. Temperature program: initial temperature, 45 °C; ramp rate, 15 °C/min to 400 °C; final temperature, 400 °C hold for 20 min.

- (3) Figures S6 in Supplementary Information on pages 13:

Figure S6. Chromatogram of the evolved gas at 400 °C during PCN-NaCA-2 synthesis process monitored by thermal gravimetric analyzer-gas chromatograph-mass spectrometer (TGA-GC-MS).

(4) Figures S7 in Supplementary Information on pages S14:

Figure S7. The mass spectra of every retention peaks in the chromatogram in Figure S6.

(5) Figures S8 in Supplementary Information on pages S15:

Figure S8. Proposed reaction mechanism of the cyanamate moiety formation based on the experimentally confirmed species (S_n , CS_2 , and SCNH) in TG-GCMS characterization as well as the discussions from Sattler and Schnick.

Comments:

“2. One weak point of this study is the use of glycerol as a hole scavenger. Glycerol is not such a cheap reagent. I think that other oxidation reaction (water splitting, pollutant degradation, organic molecule oxidation to valuable reagents, etc.) should be studied as an alternative.”

Responses:

- (1) In the revised manuscript, we discussed the status of the glycerol production in the growing bio-diesel industry. The glycerol by-product is surplus, and the utilization of this biomass derivative is important.
- (2) We choose glycerol as the electron/proton donor also for the following advantages: (1) glycerol oxidation by photo-induced holes presents fast reaction kinetics, efficiently supplying electrons/protons to the selective oxygen reduction reaction; (2) glycerol and its oxidation products are non-toxic and easily bio-degradable, and we propose the utilization of produced H_2O_2 aqueous

solution containing glycerol oxidation intermediates directly in the environmental remediation, such as the reductive elimination of the highly toxic hexavalent Cr (Figure S20).

- (3) In anaerobic reaction (photocatalytic H₂ evolution), PCN-NaCA-2 shows a protons reduction performance that is similar to PCN (Figure S25); while in the aerobic reaction with bisphenol A (BPA) as the electron/protons donor, both the H₂O₂ production rate and the BPA degradation rate on PCN-NaCA-2 are around five times of that on PCN (Figure S24); in the presence of glycerol as the electron/proton donor, the H₂O₂ production performance on PCN-NaCA-2 is more than 10 times of that on PCN. In the flow-reactor under sufficient photons irradiation and with glycerol as the proton/electron donor, PCN-NaCA-2 presents superior H₂O₂ production activity, which is 24.6 times of that on PCN. These experiments indicate that the potential of the unique surface-active sites for 2e⁻ ORR on PCN-NaCA-2 could be fully developed on condition that sufficient electrons/protons are available.

Revisions:

- (1) Revisions in the manuscript on page 5:

Developing a proper process of consuming and valorizing glycerol well matches the market need. The crude glycerol is cheap with price of 10 – 15 c/lb. (80 wt.%, Oleoline), while H₂O₂ is a moderately valuable chemical with price of around 6.0 USD/lb. (35 wt. %, Supleco).

- (2) Revisions in the manuscript on page 7:

The photocatalytically produced H₂O₂ aqueous solution is an environmental benign and efficient reductant for elimination of the highly-toxic hexavalent Cr (Figure S20).

- (3) Revisions in the manuscript on pages 7-8:

The distinction between the photocatalytic performances of PCN and PCN-NaCA-2 varies with the reaction conditions. In the flow-reactor under sufficient photons irradiation and with glycerol as the proton/electron donor, PCN-NaCA-2 presents superior H₂O₂ production activity, which is 24.6 times of that on PCN. In the photocatalytic H₂O₂ production coupled with BPA (Bisphenol A) degradation, the initial H₂O₂ production rate and BPA degradation reaction rate constant on PCN-NaCA-2 are 5 times of those on PCN (Figure S24); While in the anaerobic H₂ evolution reaction, PCN-NaCA-2 and PCN exhibit very close photocatalytic activity in terms of H₂ production rate (Figure S25). It is proposed that PCN-NaCA-2 possesses unique surface-active sites for efficient outputting of the photo-induced electrons to surface adsorbed dioxygen, and its potential for H₂O₂ production can be maximized in the presence of sufficient supply of electron/proton (Figure S25).

- (4) Figure S20 and notes in Supplementary Information on page S23:

Figure S20. Reductive elimination of Cr(VI) by photocatalytically generated hydrogen peroxide. Reaction conditions: 1.8 mL H₂O₂ solution with concentration of 10 mmol produced by PCN-NaCA-2 photocatalysis was charged into a silica cuvette. 200 uL of Cr(VI) solution with concentration of 400 ppm was added in the cuvette and mixed by magnetic stirrer. The concentration of Cr(VI) was determined by the absorbance of the solution at 340 nm with a calibration curve.

Notes: Reductive conversion of the highly toxic hexavalent Cr to less toxic trivalent Cr is one of the most important approaches for treating hexavalent Cr contamination. Hydrogen peroxide is an efficient and environmental-benign reductant for hexavalent Cr conversion as shown in the following chemical equation:

The photocatalytically generated H₂O₂ solution was applied in the Cr(VI) reduction reaction. Cr(VI) with a concentration of 40 ppm was eliminated within 3 min. Addition of Cr(VI) stock solution in to the reaction system led to a spike in Cr(IV) concentration to 40 ppm, and 5 min reaction reached 95% Cr(VI) removal. After the third spike in Cr(VI) concentration to 40 ppm, 5 min reaction reached 72% Cr(VI) removal. These tests implicate the potential application of the photocatalytically generated H₂O₂ solution for the environmental Cr(VI) pollution elimination.

(5) Figure S24 in Supplementary Information on page S26:

Figure S24. Photocatalytic H_2O_2 production coupled with BPA (Bisphenol A) degradation; (b) pseudo first-order kinetic plots of the photocatalytic BPA degradation. Reaction conditions: 10 mg photocatalyst was dispersed in 50 mL BPA aqueous solution with concentration of 100 ppm by ultrasonication. The photocatalytic reaction was conducted in a jacketed-photoreactor. The light source was a solar simulator with intensity of 100 mW/cm^2 on the surface of the jacketed reactor.

(6) Figure S25 in Supplementary Information on page S27:

Figure S25. Comparison of PCN and PCN-NaCA-2 in different reactions. (a) Photocatalytic H_2O_2 production in the flow reactor under sufficient photons irradiation in the presence of glycerol as the proton/electron donor; (b) photocatalytic H_2O_2 production in the batch reactor in the presence of glycerol as the proton/electron donor; (c) photocatalytic H_2O_2 production in the batch reactor in the presence of bisphenol A (BPA) as the proton/electron donor; (d) photocatalytic H_2 evolution in anaerobic condition.

Comments:

“3. How much Na^+ do they have in the final materials? Is it within the CN framework of more on the surface of the material?”

Responses:

- (1) The content of Na in PCN-NaCA-2 is 6.9 wt. % by ICP-OES analysis, while when it was dispersed in the 50 mL solution, the sodium ion concentration in solution is only 4.2 ppm (sodium distribution: in catalyst / in solution = 85 / 15).
- (2) The chemical environment of nitrogen and sodium in PCN-NaCA-2 was investigated with x-ray adsorption spectroscopy (XAS). It is experimentally and theoretically evidenced that the sodium is coordinatively interacting with four pyridinic nitrogen from two adjacent tri-s-triazine units. All this

indicates that the sodium cation is chemically bonded within the framework, rather than just physically adsorbed on the surface. This part is discussed in the revised manuscript.

Revisions:

(1) Revisions in the manuscript on pages 3 – 4:

The sodium content of PCN-NaCA-2 was determined to be 6.9 wt.% by inductively coupled plasma atomic emission spectroscopy (ICP OES). For further understanding the chemical environment of sodium in the framework, PCN-NaCA framework was simulated with density functional theory (DFT) based on a linear melon structure with infinite repeating units; and in the optimized configuration, there is interaction between the sodium ion and four pyridinic nitrogen atoms of two adjacent heptazine units (Figures 2b, S9). Nitrogen K-edge X-ray near-edge structure (XANES) measurements were thereafter used to investigate the interaction (Figure 2c). PCN shows typical π^* resonance at 399.3 eV corresponding to pyridinic N of the tri-s-triazine moiety;^{42,43} most importantly, blue shift and enhancement of the pyridinic N peak is observed, demonstrating the presence of coordination interaction and charge transfer from pyridinic nitrogen to sodium.^{44,45} Meanwhile, there is no shift for the other π^* resonances (401.2 eV, amino; 402.3 eV, graphitic),^{46,47} indicating no interaction between these nitrogen atoms and sodium. Furthermore, the chemical environment of the sodium was characterized with X-ray absorption spectrometer (Figure S10). Sodium in PCN-NaCA-2 presents a K-edge absorption profile that is different from that of NaSCN and NaSCN/PCN, but is very similar to the profile of sodium diformylamine. The XAS results are indicating that the sodium ion is strongly interacting with PCN-NaCA-2 framework via the coordination with pyridinic nitrogen, which matches well with the theoretical simulation.

References:

42. Zheng, Y. *et al.* Hydrogen evolution by a metal-free electrocatalyst. *Nat. Commun.* **5**, 3783 (2014).
43. Zhang, J.-R. *et al.* Accurate K-edge X-ray photoelectron and absorption spectra of g-C₃N₄ nanosheets by first-principles simulations and reinterpretations. *Phys. Chem. Chem. Phys.* **21**, 22819-22830 (2019).
44. Liang, Y. *et al.* Covalent Hybrid of Spinel Manganese–Cobalt Oxide and Graphene as Advanced Oxygen Reduction Electrocatalysts. *J. Am. Chem. Soc.* **134**, 3517-3523 (2012).
45. Lee, J. H. *et al.* Carbon dioxide mediated, reversible chemical hydrogen storage using a Pd nanocatalyst supported on mesoporous graphitic carbon nitride. *J. Mater. Chem. A* **2**, 9490-9495 (2014).
46. Leinweber, P. *et al.* Nitrogen K-edge XANES - an overview of reference compounds used to identify 'unknown' organic nitrogen in environmental samples. *J. Synchrotron Radiat.* **14**, 500-511 (2007).

47. Ripalda, J. M. et al. Correlation of x-ray absorption and x-ray photoemission spectroscopies in amorphous carbon nitride. *Phys. Rev. B* **60**, R3705-R3708 (1999).

(2) Revised Figure 2 in the manuscript on page 4:

Figure 2. Synthesis and structure of the photocatalysts. (a) Synthesis of PCN and PCN-NaCA-n. (b) The optimized configurations of PCN and PCN-NaCA by DFT calculation. (c) N K-edge XANES of PCN and PCN-NaCA-2. The insets show enlarged peaks of pyridinic nitrogen and graphitic nitrogen.

(3) Figure S10 in Supplementary Information on page S16:

Figure S10 Na K-edge X-ray absorption spectra. (a) NaSCN (20 % w.t.) / PCN prepared via Incipient wetness impregnation; (b) Sodium thiocyanate; (c) PCN-NaCA-2; (d) Sodium diformylamine.

Comments:

“4. A comparison to PCN after heat treatment at 400 C should be given as well. In addition, how can the authors exclude the creation of N vacancies after post-heating to 400 C (as shown many times before)? The latter may be the reason for the absorption shift and enhanced catalytic activity.”

Responses:

- (1) PCN was prepared under 600 °C in nitrogen atmosphere. Additional 3 hours heat treatment under 400 °C in nitrogen flow is unable to obviously change the surface properties.
- (2) Following the reviewer’s suggestions, we treated PCN under 400 °C for 3 hours (PCN-400). As shown in Figure 1a, PCN-400 and PCN were employed as the control samples, and they exhibited the same performance in the photocatalytic H₂O₂ generation.

Revisions:

- (1) Revisions in the manuscript on page 5:

PCN and PCN-400 (PCN heated in nitrogen atmosphere at 400 °C for 3 hours) exhibit the same photocatalytic performance, generating 0.23 mM H₂O₂ in 45 min irradiation, ...

- (2) Revised Figure 3b in the manuscript on page 6:

Comments:

“5. Long-term stability measurements are missing, including post-characterization and recycling of the catalyst.”

Responses:

- (1) Long-term photocatalytic reaction was conducted. As shown in Figure S17, H₂O₂ concentration increases with the irradiation time for 10 hours, and H₂O₂ concentration reached 11.1 mM. The photocatalyst was separated for post-characterization.
- (2) The cycle performance of PCN-NaCA-2 in the photocatalytic H₂O₂ production reaction was examined. As shown in Figure S16, the 8th recycled photocatalyst still exhibited good photocatalytic performance that is 88.1 % of initial run.
- (3) The sodium content of the photocatalyst after 10 hours reaction was determined to be 1.8 wt. % by ICP-OES.
- (4) PCN-NaCA-2 after 10 h of photocatalytic reaction was collected and characterized by XPS. The changes in N1s and C1s signal was discussed in the revised supporting information.

Revisions:

- (1) Revisions in the manuscript on pages 5-6:

PCN-NaCA-2 shows stable performance for recycling, and at 8th run, the photocatalytic performance is 88.1% of the initial run (Figure S16). Long-term running stability of PCN-NaCA-2 was examined in a batch reaction, and H₂O₂ concentration reaches a remarkable value of 11.1 mM after 10 h

irradiation (Figure S17). The characterizations of the recycled photocatalyst indicate that long-term photocatalytic reaction leads the loss of the amine moiety as well as sodium ion in the framework (Figure S18).

(2) Figure S16 in Supplementary Information on page S20:

Figure S16. Cycle test of PCN-NaCA-2 in the photocatalytic H₂O₂ production. Reaction conditions: 20 mg PCN-NaCA-2 was added in the 50 mL aqueous solution with 3.5 wt.% glycerol in a 50 mL jacketed photo-reactor at room temperature of 25 °C. The reaction mixture was irradiated by solar simulator for 30 min for each cycle. After the reaction, the photocatalyst was separated by centrifugation, and 50 mL fresh aqueous glycerol solution (3.5 wt. %) was charged into the photoreactor with the recycled photocatalyst for the next run.

(3) Figure S17 in Supplementary Information on page S21:

Figure S17. Long term running performance of PCN-NaCA-2 in photocatalytic H₂O₂ production. Reaction conditions: 20 mg photocatalyst was dispersed in 50 mL aqueous solution with glycerol contents of 3.5 wt. % was charged in the photoreactor and irradiated with solar simulator.

(4) Figure S18 and notes in Supplementary Information on page S21:

Figure S18. XPS characterization of PCN-NaCA-2 recycled after 10 hours photocatalytic H_2O_2 production.

Notes: for N1s signal, the deconvoluted peaks at 400.1 eV decreases as compared to that of as-prepared PCN-NaCA-2, which might result from the changes in the amine moiety of PCN-NaCA-2 after long-term running under irradiation; for C1s signal, there is no obvious changes except that the increased peak intensity at 284.9 eV, which could be attributed to the surface deposited adventitious carbon after the reactions involving organics. The sodium content after long-term irradiation was determined to be 1.8 wt.% by ICP-OES.

Comments:

“6. The authors state that the absorption is better in the range of 450 – 500 nm. However, according to the AQE, there is no activity in this region. What are the reasons for that? Usually, this enhanced absorption can be attributed to defects states below the conduction band.”

Responses:

We appreciate the reviewer for the suggestions. We agree with the reviewer and revised the discussions on absorption spectra.

Revisions:

Revisions in the manuscript on page 9:

The absorbance at 450 – 500 nm might result from the excitation to the defects states below the conduction band.

Comments:

“7. How can they avoid the oxidation of Glycerol by hydroxyl radicals? It may inhibit that reaction after a short time as the peroxide will be consumed.”

Responses:

- (1) As shown in Figure S15., the photocatalytic H_2O_2 decomposition reaction was found to be very slow.
- (2) Glycerol plays the role of electron/proton donor for the H_2O_2 production, and what we are expecting is the efficiency of H_2O_2 production; the oxidative degradation of the glycerol is not concerned in this work.
- (3) Moreover, the selectivity of electron/proton towards the formation of H_2O_2 production is determined to be >93% (Figure S23), i.e., in this photo-redox reaction, O_2 to H_2O_2 is predominant in the reduction side and glycerol oxidation by hole is the majority in the oxidation side.

Revisions:

- (1) Revisions in the manuscript on page 5:

The hydrogen peroxide decomposes slowly on PCN-NaCA-2 and PCN (Figure S15), which contributes positively to the H_2O_2 accumulation in the reaction system.

- (2) Figure S15 in Supplementary Information on page S20:

Figure S15. Photocatalytic decomposition of H_2O_2 on PCN and PCN-NaCA-2 in the presence (a) and in the absence of glycerol (3.5 wt. %). Reaction conditions: 5 mg photocatalyst and 35 mL H_2O_2 solution with concentration of 6.5 mM was charged in a 50 mL photoreactor. The photoreactor was sealed and the oxygen in the reactor was removed by repeated vacuum and nitrogen refilling. The oxygen free reaction mixture was then irradiated with solar simulator.

Comments:

“8. *The mechanism of glycerol degradation should be studied.*”

Responses:

The degradation mechanism of the glycerol in the photocatalytic reaction was studied *via* intermediates analysis. Based on various identified intermediates, we proposed the mechanism of the glycerol degradation (Figure S22).

Revisions:

(1) Revisions in the manuscript on page 7:

The biomass-derived glycerol serves as the electron/proton donor for O₂ to H₂O₂ conversion. The glycerol degradation process was investigated *via* reaction intermediates identification. By gas chromatography-mass spectrometry (GC-MS), four major intermediates, e.g., dihydroxyacetone, glyceraldehyde, glyceric acid, and glycolic acid, are identified. Based on these major intermediates, the glycerol degradation process by hole oxidation is proposed accordingly (Figure S22).

(2) Figure S22 in Supplementary Information on page S25:

Figure S22. Scheme of the proposed reaction mechanism of dioxygen reduction and glycerol degradation. The glycerol oxidation reaction mechanism was proposed based on the GC-MS identified intermediates, e.g., dihydroxyacetone, glyceraldehyde, glyceric acid, and glycolic acid.

Comments:

“9. A summary of the outcome of the TAS and SPV will be useful for the reader (also as an illustration).”

Responses:

In this work, fs-TAS shows the dynamics of the trapped electrons; and SPV measurements reveal the

information on the amount of charge carriers reaching the surface. They are consistent in demonstrating that the construction of the sodium cyanamate moiety significantly improves the charge separation efficiency. Considering the constructive comments, in the revised manuscript, there is a short summary on these results.

Revisions:

Revisions in the manuscript on page 12:

As shown in Figure 7a, PCN presents very weak positive photovoltage at the band of 300 – 350 nm; while PCN-NaCA-2 presents an intensive negative SPV signal at the band of 300 - 400 nm, demonstrating the significantly enhancement of charge separation efficiency on PCN-NaCA-2 than that on PCN.⁸⁰⁻⁸⁴ The steady-state observation here is consistent with fs-TAS results, in which the amount of shallow trapped electron produced by laser pulse on PCN-NaCA-2 is significantly higher than that on PCN.

References:

80. Jiang, T. et al. Photoinduced charge transfer in ZnO/Cu₂O heterostructure films studied by surface photovoltage technique. *Phys. Chem. Chem. Phys.* **12**, 15476-15481 (2010).
81. Gross, D. et al. Charge separation in type II tunneling multilayered structures of CdTe and CdSe nanocrystals directly proven by surface photovoltage spectroscopy. *J. Am. Chem. Soc.* **132**, 5981-5983 (2010).
82. Wei, X. et al. Effect of heterojunction on the behavior of photogenerated charges in Fe₃O₄@Fe₂O₃ nanoparticle photocatalysts. *J. Phys. Chem. C* **115**, 8637-8642 (2011).
83. Townsend, T. K., Browning, N. D. & Osterloh, F. E. Overall photocatalytic water splitting with NiO_x-SrTiO₃ - a revised mechanism. *Energy Environ. Sci.* **5**, 9543-9550 (2012).
84. Lei, B. et al. In situ synthesis of α-Fe₂O₃/Fe₃O₄ heterojunction photoanode via fast flame annealing for enhanced charge separation and water oxidation. *ACS Appl. Mater. Interfaces* **13**, 4785-4795 (2021)

Comments:

“10. The title is slightly misleading – in this paper the authors don't really identify the factor for...”

Responses:

A few words in the title has been revised.

Revisions:

Revision on the title of the manuscript:

Mechanistic Analysis of Multiple Processes Controlling Solar-driven H₂O₂ Synthesis Using Engineered Polymeric Carbon Nitride

Reviewer #4 (Remarks to the Author):

Comments:

"This study reported a molten salt method that can introduce sodium cyanamate (NaCA) moiety to polymeric carbon nitride (PCN) frameworks. The authors then investigated the effect and mechanisms of NaCA of PCN on the photochemical synthesis of H₂O₂ in glycerol solution under simulated solar light. I would recommend it for publication in Nature Communications after the following critical issues are solved."

Responses:

We appreciate the reviewer for the following constructive suggestions, and we have carefully designed experiments and addresses all the concerns, and made detailed revisions on the manuscript and Supplementary Information.

Comments:

"1. Fig.S1e the NMR of PCN-NaAC-2 did not show any obvious differences compared with that of the PCN sample. Can the author enlarge the specific part to show the peak at 171 ppm?"

Responses:

The shoulder peak at 171 ppm has been enlarged in Figure S1.

Revisions:

Revised Figure S1e in Supplementary Information on page S8:

Comments:

“2. The author used 0.38M glycerol aqueous solution for H₂O₂ production and claimed the glycerol worked as electron and proton donors.”

I recommend the analysis of the final oxidation product of glycerol to get solid evidence for this conclusion. For instance, how much of the raw glycerol is oxidized to glyceraldehyde after the photochemical reaction? Can the concentration of glycerol be lower considering that 0.38M is quite high?”

Responses:

- (1) As shown in the glycerol oxidation reaction mechanism analysis, there are various intermediates, which are including $2e^-/2H^+$, $4e^-/4H^+$, and more electrons/protons transfer reaction products. We thus examined the selectivity of proton/electron towards O₂ to H₂O₂ conversion with 4-methylbenzyl alcohol as the electron/proton donor, due to the accuracy in quantifying the $2e^-$ oxidation product (4-methylbenzaldehyde). The selectivity of electron/proton towards the formation of H₂O₂ production is determined to be >93% (Figure S23), i.e., in this photo-redox reaction, alcohol oxidation is supplying proton/electron to the conversion of O₂ to H₂O₂.
- (2) The impact of the glycerol concentration on the hydrogen peroxide generation performance was examined and discussed in the Supporting Information. As shown in Figure S11, the H₂O₂ production performance increases with the glycerol concentration; the impact becomes mild when the glycerol concentration is higher than 3.5 wt. % (0.38 M). For a detailed mechanistic investigation on the photocatalysts, we evaluated the catalytic performance in the presence of sufficient electron/proton

donor (0.38 M / 3.5 wt. % glycerol).

Revisions:

(1) Revisions in the manuscript at page 7:

As there are multiple intermediates/products in glycerol oxidation reaction, selective oxidation of 4-methylbenzyl alcohol to 4-methylbenzaldehyde was employed for further examining the H₂O₂ selectivity. As shown in Figure S23, H₂O₂ production selectivity (in terms of the molar ratio between H₂O₂ and 4-methylbenzaldehyde) reaches > 93%, stating that the alcohol to aldehyde conversion is supplying electrons/protons to the dioxygen reduction for H₂O₂ production.⁶⁵

Reference:

65. Krishnaraj, C. *et al.* Strongly reducing (diarylamino)benzene-based covalent organic framework for metal-free visible light photocatalytic H₂O₂ generation. *J. Am. Chem. Soc.* **142**, 20107-20116 (2020)

(2) Figure S23 in Supplementary Information at page S26:

Figure S23. Photocatalytic H₂O₂ production with 4-methylbenzyl alcohol as the electron/proton donor on PCN-NaCA-2 under 420 nm irradiation. Reaction conditions: 20 mg PCN-NaCA-2 was dispersed in 10 mL acetonitrile with 1.5 g 4-methylbenzyl alcohol and 0.15 g H₂O. The photoreactor was filled with 1 atm. O₂, capped, and irradiated by a LED lamp ($\lambda = 420 \text{ nm}$, 25.0 mW/cm²). 4-Methylbenzaldehyde was quantified by GCMS with calibration curve and 1,4-dicyanobenzene as the internal standards.

(3) Figure S11 in Supplementary Information on page S17:

Figure S11. Photocatalytic H₂O₂ production performance of PCN-NaCA-2 in the aqueous solution with various glycerol contents. Reaction conditions: 10 mg photocatalyst and 50 mL aqueous solution with various glycerol contents was charged in the photoreactor and irradiated with solar simulator for 1 hour.

Comments:

“3. Following the previous question, the use of glycerol as sacrifice agent would increase the cost and bring impurities to the final H₂O₂ production. Glycerol itself is a useful chemical. Do the authors think that the produced H₂O₂ will be more valuable? Besides, how to purify the H₂O₂ from the final mixture for further usage?”

Responses:

- (1) In the rising biodiesel industry, especially in Europe Union, glycerol is the byproduct and its yield accounts for 10 wt.% of the biodiesel production (<https://www.ebb-eu.org/stats.php>). Although glycerol is a versatile chemical, the increasing production and limited consumption makes glycerol surplus. To develop the utilization of glycerol is quite important.
- (2) The crude glycerol is cheap with price of 10 – 15 c/lb. (80 wt.%, Oleoline), while H₂O₂ is a moderately valuable chemical with price of around 6.0 USD/lb. (35 wt. %, Supleco). In the revised manuscript, we briefly discussed the comparison of glycerol and hydrogen peroxide from a commercial perspective.
- (3) To purify hydrogen peroxide from the final mixture is indeed not easy. However, we are considering the application of hydrogen peroxide in environmental remediation. The H₂O₂ containing mixture can be used for elimination of the environmental pollutants without purification, as the glycerol and its oxidative products are easily bio-degradable, and are not identified as environmental pollutants.
- (4) In the revised supporting information, we demonstrated this concept by employing the final mixture

with 10 mM H₂O₂ in the reductive elimination of Cr(VI). As shown in Figure S20, the mixture is capable of eliminate Cr(VI) efficiently.

Revisions:

(1) Revision in the manuscript on page 5:

Developing a proper process of consuming and valorizing glycerol well matches the market need. The crude glycerol is cheap with price of 10 – 15 c/lb. (80 wt.%, Oleoline), while H₂O₂ is a moderately valuable chemical with price of around 6.0 USD/lb. (35 wt. %, Supleco).

(2) Revision in the manuscript on page 7:

The photocatalytically produced H₂O₂ aqueous solution is an environmental benign and efficient reductant for elimination of the highly-toxic hexavalent Cr (Figure S20).

(3) Figure S20 and notes in Supplementary Information on page S23:

Figure S20. Reductive elimination of Cr(VI) by photocatalytically generated hydrogen peroxide. Reaction conditions: 1.8 mL H₂O₂ solution with concentration of 10 mmol produced by PCN-NaCA-2 photocatalysis was charged into a silica cuvette. 200 uL of Cr(VI) solution with concentration of 400 ppm was added in the cuvette and mixed by magnetic stirrer. The concentration of Cr(VI) was determined by the absorbance of the solution at 340 nm with a calibration curve.

Notes: Reductive conversion of the highly toxic hexavalent Cr to less toxic trivalent Cr is one of the most important approaches for treating hexavalent Cr contamination. Hydrogen peroxide is an efficient and environmental-benign reductant for hexavalent Cr conversion:

The photocatalytically generated H₂O₂ solution was applied in the Cr(VI) reduction reaction. Cr(VI) with a concentration of 40 ppm was eliminated within 3 min. Addition of Cr(VI) stock solution in to the

reaction system led to a spike in Cr(IV) concentration to 40 ppm, and 5 min reaction reached 95% Cr(VI) removal. After the third spike in Cr(VI) concentration to 40 ppm, 5 min reaction reached 72% Cr(VI) removal. These tests implicate the potential application of the photocatalytically generated H₂O₂ solution for the environmental Cr(VI) pollution elimination.

Comments:

“4. The H₂O₂ decomposition also happens in photocatalysis (e.g. oxidized by holes) and in alkaline solution. The authors are suggested to conducted experiments to investigate the decomposition rates of PCN and PCN-NaAC-2.”

Responses:

In the revised manuscript, we analyzed the photocatalytic H₂O₂ decomposition behavior on PCN and PCN-NaCA-2. As shown in Figure S15, 2 hours light irradiation in the absence of dioxygen led to less than 9 % decrease in the H₂O₂ concentration for both PCN and PCN-NaCA-2, and PCN-NaCA-2 exhibited slightly faster degradation rate than PCN.

Revisions:

(1) Revisions in the manuscript on page 6:

The hydrogen peroxide decomposes slowly on PCN-NaCA-2 and PCN (Figure S15), which contribute positively to the H₂O₂ accumulation in the reaction system.

(2) Figure S15 in Supplementary Information on page S20:

Figure S15. Photocatalytic decomposition of H₂O₂ on PCN and PCN-NaCA-2 in the presence (a) and in the absence of glycerol (3.5 wt. %). Reaction conditions: 5 mg photocatalyst and 35 mL H₂O₂ solution with concentration of 6.5 mM was charged in a 50 mL photoreactor. The photoreactor was sealed and the oxygen in the reactor was removed by repeated vacuum and nitrogen refilling. The

oxygen free reaction mixture was then irradiated with solar simulator.

Comments:

“5. Line 102. What is the flow rate in the continuous serial micro-batch reactor? More details should be provided in the SI, such as catalyst suspension flow rate, O₂ flow rate and the light wavelength and intensity of column LED array?”

Responses:

More details of the continuous serial micro-batch reactor have been added in the revised manuscript in the METHODS section.

Revisions:

Revision in the manuscript on page 16:

Gaseous oxygen and photocatalyst suspension were fed at the same flow rate of 1.3 mL/min for retention time of 36 min in the flow reactor; and setting the flow rate of 0.65 mL/min affords a retention time of 72 min. A column LED array (Household white light LED column) is placed coaxially inside the coil reactor, and the light intensity at the surface of the PTFE tubing was 27 mW/cm².

Comments:

“6. Line 249-260. On the 2e⁻ ORR for H₂O₂ production, it can further be classified to one-step direct 2e⁻ ORR and two-step indirect 2e⁻ ORR (forming superoxide radical first) by PCN. According to previous studies (e.g. Chem. Commun., 2019, 55, 13279-13282; ACS Catal., 2020, 10, 3697-3706), the indirect 2e⁻ ORR process can also promote the generation of H₂O₂ production. The authors need to further conduct some experiments (For example EPR spectroscopy) to see whether there were superoxide anions generated in photocatalysis. This analysis can further understand the photocatalysis reaction mechanism.”

Responses :

- (1) By employing EPR technique, we have qualitatively confirmed the presence of superoxide radicals (Figure S21a).
- (2) To evaluate the contribution of the superoxide route to the H₂O₂ formation, superoxide dismutase (SOD) was used for accelerating the superoxide radicals dismutation reaction for H₂O₂ production. As shown in Figure S21b., the addition of SOD does not obviously change H₂O₂ production rate, i.e., superoxide dismutation reaction contributes negligible to the overall H₂O₂ production, and one-step 2e transfer is the predominant pathway.

Revisions:

- (1) Revisions in the manuscript on page 7:

There are two pathways for H₂O₂ production, e.g., one-step two electrons transfer pathway and superoxide radicals involved pathway. For analyzing the H₂O₂ formation mechanism, the contribution of each pathway in the reaction system was evaluated. As shown in the electron spin resonance (ESR) spectra (Figure S21a), the peak intensity of DMPO (5,5-dimethyl-1-pyrroline N-oxide)-superoxide radical adduct, although being weak, increases with irradiation time, stating the photocatalytic production of superoxide radicals. Superoxide dismutase (SOD) can efficiently accelerate the superoxide radical dismutation reaction towards H₂O₂ production in both biological and photocatalytic reaction systems.⁶²⁻⁶⁴ For evaluating the contribution of the superoxide involved pathway to the overall H₂O₂ production, SOD was added in PCN-NaCA-2 photocatalyzed reaction system. However, with the increase of SOD concentration, there is no obvious acceleration of the H₂O₂ production observed (Figure S21b). This probe reaction is demonstrating that the superoxide radicals involved pathways contribute negligibly to the overall H₂O₂ production.

References:

62. Sheng, Y. *et al.* Superoxide dismutases and superoxide reductases. *Chem. Rev.* **114**, 3854-3918 (2014)
63. Waiskopf, N. *et al.* Photocatalytic reactive oxygen species formation by semiconductor-metal hybrid nanoparticles. toward light-induced modulation of biological processes. *Nano Lett.* **16**, 4266-4273 (2016).
64. Qian, Y., Li, D., Han, Y. & Jiang, H.-L. Photocatalytic molecular oxygen activation by regulating excitonic effects in covalent organic frameworks. *J. Am. Chem. Soc.* **142**, 20763-20771 (2020).

- (2) Figure S21 in Supplementary Information on page S24:

Figure S21. (a) DMPO (5,5-Dimethyl-1-pyrroline N-oxide) – superoxide radical adduct signal monitored by ESR (Electron Spin Resonance) at various reaction times on PCN-NaCA-2 in oxygen saturated $\text{CH}_3\text{CN}/\text{CH}_3\text{OH}$ (v/v, 10/1) solvent; (b) Examination of the contribution of superoxide radical disproportionation on the H_2O_2 production by SOD (superoxide dismutase) probe reaction. Reaction conditions: PCN-NaCA-2 (0.2 mg/mL), 2 mL phosphate buffer solution (1 mM, pH 7.4) with 3.5 wt. % glycerol and various SOD concentration was charged in a photoreactor; the oxygen-saturated reaction mixture was irradiated with 420 nm LED for 1 min and then sampled for H_2O_2 concentration analysis.

Comments:

“7. Line 269. Can the author explain why the theoretical simulation of $2e^-$ ORR steps were performed in alkaline solution, but not acid or neutral solution?”

Responses:

PCN-NaCA-2 shows optimum photocatalytic H_2O_2 performance in weak basic conditions, and $2e^-$ ORR selectivity measurements with RRDE was also conducted in the NaOH solution. We thus performed the theoretical simulation in the alkaline condition.

Reviewer #5 (Remarks to the Author):

Comments:

“This is a detailed investigation (mostly experimental, with some calculations) on the photocatalytic evolution of H_2O_2 using polymeric carbon nitride (PCN) based photocatalysts that have been treated with NaSCN in order to add sodium cyanamine moieties. The paper shows an enhancement of H_2O_2 evolution over non-treated PCN materials. The enhancement is quantified by a table in the SI, showing an

approximately sixfold increase in H₂O₂ production rate over the past attempts show, a list which is restricted to different varieties of PCN.

I have a variety of questions and criticisms of the paper. Some are fundamental and some are technical.

1) I do not see a clearly quantified reason why this study reflects a high-profile advance (Nature Communications) rather than a contribution to the chemical literature that is, while valuable, essentially a technical report. It is not clear that the reported sixfold increase will make a difference between what is an interesting technical observation and a true advance that would (as indicated in the introduction) sway industry away from its conventional processes. How does the process compare to other processes in the literature? The substances used/synthesized here are not new, so there is no new understanding of the fundamental processes and/or components of the system either.”

Responses:

- (1) We appreciate the reviewer for very detailed and constructive comments; and the revisions based on these comments substantially improve the manuscript.
- (2) After an intensive revision, we would like to briefly emphasize the background and the significance of this work.

There is no doubt that solar energy conversion is of great importance for a sustainable energy future. The gap between solar energy potentials and the practical application of it is the cost-effectiveness of the solar conversion systems. The cost-effectiveness relies on the low cost and high efficiency of the materials on which the photon energy is converted/stored. Low cost always related to the abundance of the composition and the fabrication processes. Polymeric carbon nitride materials, composed of earth abundant elements, can be fabricated at very low cost. However, the solar energy conversion efficiency is so far unsatisfactory. The efficiency improvement could be enabled by controlling the photocatalyst in nanoscale, which must be based on a comprehensive understanding of the fundamental mechanisms.

- (3) In this work, we are dedicated to revealing the rationale behind molecular structure-efficiency relationship in a superior photocatalytic H₂O₂ production system. All the events, from photons excitation step to the final H₂O₂ molecule formation step, are systematically examined, during which some previously unknown structural features and interactions are experimentally substantiated.
 - ◇ As described in the reviewer’s comments, the detailed structures of various polymeric carbon nitride materials are so far controversial, due to the lack of substantiated structural information. In this work, the sodium in the carbon nitride framework, was evidenced to be interacting with four pyridinic nitrogen from two adjacent tri-s-triazine units by both XANES characterizations and theoretical simulations, which greatly promotes our understanding on the nature of polymeric carbon nitride materials.

- ✧ Most importantly, we experimentally reveal a previously unknown electronic coupling interaction on the O₂ adsorbed surface under photons excitation. The interaction significantly boosts the quantity and prolongs the lifetime of the active shallow-trapped electrons, which is highly desirable for an efficient solar conversion process.
- ✧ The findings could arouse widespread interest in considering the reactant-surface interactions during photons excitation in a variety of photocatalytic reaction systems, such as CO₂ reduction, N₂ reduction, organics conversion, et al. It is providing the solar conversion research community with a new perspective, from which some new mechanistic insights could be revealed. Meanwhile, the outcomes are very informative for rational design of materials for efficient solar-to-chemical conversion processes.

Revisions:

(1) Revisions in the Abstract:

The overall photocatalytic transformation process is systematically analyzed, and some previously unknown structural features and interactions are substantiated via experimental and theoretical methods. The structural features of cyanamino group and pyridinic nitrogen-coordinated sodium in PCN-NaCA promote photon absorption, alter the energy landscape of the framework and improve charge separation efficiency, enhance surface adsorption of dioxygen, and create highly selective 2e⁻ ORR surface-active sites. Particularly, a previously unknown electronic coupling interaction of O₂ and surface, which significantly boosts the population and prolongs the lifetime of the active shallow-trapped electrons, is experimentally substantiated.

(2) Revisions in Introduction on page 1:

However, the gap between the potentials of solar energy and practical application of it is the cost-effectiveness. Moreover, the earth-abundant photocatalyst composition and high efficiency of photons to chemical conversion process are two critical factors for cost-effectiveness.^{19,20} Therefore, design of a low-cost photocatalyst with decent solar conversion efficiency is the major challenge to the goal of sustainable H₂O₂ production. It is particularly worth noting that the efficiency improvement relies heavily on the comprehensive mechanistic understanding of the structure-activity relationship in nanoscale.²⁰

References:

19. Gray, H. B. Powering the planet with solar fuel. *Nat. Chem* **1**, 7-7 (2009).
20. Crabtree, G. W. & Lewis, N. S. Solar energy conversion. *Phys. Today* **60**, 37-42 (2007).

Comments:

"2) The mechanistic interpretation of the results, while approached for multiple angles, is, ultimately, not stringent. I recognize that this is a difficulty that anyone working on PCN materials must face. The atomic structure of the actual materials in question, especially the nature of active sites, is simply not understood in detail - nor is it possible to determine this atomic structure with certainty by any existing experimental or computational techniques, to the best of my understanding. Any model must therefore remain qualitative and speculative, except for the overall observations that can be extracted directly from experiment or from theory.

But even given these difficulties, some questions remain inherently open based on the present paper."

Responses:

- (1) We appreciate the reviewer for understanding the challenges in this work.
- (2) After considering the comments from all the reviewers, we conducted intensive experimental and theoretical simulation works. The revised manuscript now reveals more substantiated structural details.

Comments:

"2a) The paper assumes that O₂ is being converted to H₂O₂ - but no evidence is given that it is indeed O₂, and not something else, that is being converted. What is the source of this O₂? What is its partial pressure / concentration? There is glycerol in this system. Why is it not glycerol that reacts and, in the process, releases H₂O₂? There are other interesting ingredients in the system (KOH, HClO₄, perhaps others) and the role of the oxygen-containing anions is not discussed.

So, evidence that it is indeed O₂ that is being converted, and not something else, should be provided. But in any case, I missed what is the O₂ source and how it is controlled. In a quantitative chemical study, it would be essential to control, characterize and report the reagents appearing on either side of the reaction."

Responses:

- (1) Oxygen was bubbled in the photocatalyst suspension, and the dissolved oxygen is converted to H₂O₂. By controlling the atmosphere, O₂ is proven to be the oxygen source of H₂O₂ production.
- (2) It is experimentally evidence that the electrons/protons from alcohol to O₂ reduction for H₂O₂

production present a selectivity of >93%, stating that the alcohol is the electron/proton donor for the O₂ reduction to produce H₂O₂.

Revisions:

(1) Revisions in the manuscript on page 5:

As expected, the oxygen partial pressure directly impacts the H₂O₂ production performance; air atmosphere lowers the H₂O₂ production, and in the nitrogen atmosphere, trace H₂O₂ is produced from the residual dissolved oxygen in water after nitrogen flushing (Figure S14).

(2) Figure S14 in Supplementary Information on page S19:

Figure S14. Comparison of the photocatalytic H₂O₂ production performance in O₂, air, and N₂ atmosphere. Reaction conditions, the same to that in figure S13, except that the atmosphere is controlled. Reaction conditions: 10 mg photocatalyst, 50 mL aqueous solution with glycerol concentration of 3.5 wt.% was mixed by ultrasonication and irradiated by solar simulator for 45 min.

Comments:

"2b) Structural assumptions made in the discussion of the material. The paper is careful in its text descriptions of PCN derived materials. In particular, it is good to see that a variety of melon, rather than a hypothetical H-free C₃N₄ material, forms the foundation of at least the computational understanding."

Nevertheless, Figure 1a shows a depiction of a hypothetical structure of PCN-NaCA-n that is, in my view, unsubstantiated in this paper and has been subject to an extensive discussion in the literature. The crux is that the C₃N₄ like structure shown here likely cannot be made and, to my knowledge, there is no firm evidence in the literature that it ever has been made.

Among the references cited (indirectly) on this point: Refs 37, 38, and 47, which are used as evidence, are early and predate this debate. In these papers, the existence of a fully C₃N₄ like condensate was essentially a plausible assumption, but later debunked. Ref. 35 is a review and Ref 36 also presents no structural evidence. Ref. 35 states specifically: "Unfortunately, the crystal structure of heptazine-based CN is still not very clear."

Furthermore, there is at least one reference (Chemistry of Materials 29 (10), 4445-4453) that shows from simple thermodynamic considerations that the C₃N₄ like structure shown

Figure 2a (right) cannot arise under plausible thermodynamic conditions. In short, I think the unsubstantiated structural hypothesis of heptazine based C₃N₄ condensation (Fig 2a right) should be avoided and a more open nanostructure (like melon/PCN, Fig 2a middle or some similar intermediate) should be the foundation of the discussion and figures.”

Responses:

We appreciate the reviewer for the constructive comments; and we have revised the scheme of the chemical structure of PCN-NaCA-n, which is now similar to PCN in linear melon structure.

Revisions:

Revised Figure 2a in the manuscript on page 4:

Comments:

“2c) Similar for “the improved polymerization degree increases layer buckling, ...” in the discussion ... there is no evidence for this in the paper.”

Responses:

Following the abovementioned revision on the proposed structure of PCN-NaCA-2, this description has been removed from the manuscript.

Comments:

"2d) I agree with much of the qualitative discussion of the experimental results (trapped charges, charge migration). One thing that remains unclear to me is the SPV spectrum in Fig 7b. The abrupt onset at 0.8 μ s is not clearly explained and the explanation in terms of charge diffusion from bulk to surface is not clear to me at all. The sharp onset does not look like diffusion, which is a gradual process. It is difficult to understand why there is some time constant here with no signal at all."

Responses:

- (1) We acknowledge the reviewer for this comment and apologies for the unclear description of the SPV/TPV experiments and data interpretation. We added some marks/notes on the TPV spectra as well as more data interpretation in the revised manuscript for promoting the reader's understanding on the data.
- (2) There are two peaks in the TPV spectra for both PCN and PCN-NaCA-2, the first peak (peak a in Figure 7b) at 0.8 μ s rising sharply and is resulting from the drift of the charge carriers. The second peak (Peak b in Figure 7b) comes from the diffusion of the charge carriers.
- (3) We have described the details of the equipment for collecting the TPV data in Supplementary Information. Due to the resolution of the equipment, the retardation time is unable to be monitored. The time constant before the peaks conveys no information related with the photo-physic process of the samples. For avoiding misunderstanding, we added a note in Figure 7b, indicating the photovoltage signal appears as soon as the incidence of laser pulse.

Revisions:

- (1) Revised Figure 7b in the manuscript on page 13:

(2) Revisions in the manuscript on page 12:

The dynamics behavior of the charge carriers is further investigated by transient photovoltage (TPV) characterization. For both PCN and PCN-NaCA-2, there are two photovoltage peaks (peak a and peak b, Figure 7b), which are, respectively, attributed to drift and diffusion of the photo-induced charges.^{84,85} PCN-NaCA-2 presents a remarkably strong and negative peak b lasting for 2.5 ms before decaying to zero, speaking for the diffusion of large number of photo-induced electrons to the surface after photons excitation event. The photovoltage (PV) characterizations demonstrate, obviously, that the depletion of the charge carriers by recombination is significantly attenuated by the construction of the sodium cyanamate moiety on polymeric carbon nitride framework.

References:

84. Mora-Seró, I., Dittrich, T., Garcia-Belmonte, G. & Bisquert, J. Determination of spatial charge separation of diffusing electrons by transient photovoltage measurements. *J. Appl. Phys.* **100**, 103705 (2006).
85. Kronik, L. & Shapira, Y. Surface photovoltage phenomena: theory, experiment, and applications. *Surf. Sci. Rep.* **37**, 1-206 (1999).

Comments:

“2e) I believe that the computational data shown are inconclusive and not well substantiated. I should prequel this by saying that I understand well that the models itself need to remain conceptual since no experimental structural evidence regarding the actual reaction site or atomic geometry exists at all, nor is it clear which statistical ensemble of reaction sites should be considered. So the conceptual models themselves are certainly necessary and the existence of the PCN sites shown is not implausible.”

Responses:

We appreciate the reviewer for the constructive comments. By taking all the comments (from all the reviewers) into consideration, we designed a couple of experiments for addressing the concerns, and the results are informative, such as the interaction between sodium and pyridinic nitrogen evidenced by XANES. Based on the updated experimental results, we constructed new models and started new theoretical simulation investigations from scratch. The new results and discussions in the simulation studies are updated in the revised manuscript.

Comments:

“However, even then, there are several unsubstantiated assumptions and technical omissions in the theory, which raise doubts:

- How plausible is the presence of Na⁺ in the site shown? This is an equilibrium with H₂O, or so I understand, and Na⁺ is rather soluble. What is the expected concentration of Na⁺ in such a site, compared to Na⁺ in solution? There are multiple other ionic species around. A thermodynamic analysis of the probability of Na⁺ occupying the sites in question should be provided. (The presence of Na⁺ is a critical prerequisite for the energetics claimed and so this assumption should be substantiated.)”

Responses:

- (1) By ICP-OES (Inductively coupled plasma atomic emission spectroscopy) analysis, PCN-NaCA-2 has a sodium content of 6.9 wt.%; and in the equilibrated aqueous suspension with PCN-NaCA-2 concentration of 0.2g/L, nearly 90 % of the sodium is distributed in PCN-NaCA-2, indicating a strong interaction between sodium and the carbon nitride framework.
- (2) The position of sodium is screened for a reasonable configuration of PCN-NaCA. Three possible positions for sodium are proposed, and the optimization processes affords the same final configuration, in which the sodium interacts with four pyridinic nitrogen from two adjacent tri-s-triazine units.
- (3) The chemical environment of sodium and nitrogen in the framework was further characterized with XANES (X-ray absorption near edge structure). In nitrogen K-edge XANES measurement, the blue shift and intensity enhancement of the π^* resonances at 399.3 eV (pyridinic N) speaks for the charge transfer and interaction between pyridinic N and Na. The Na K-edge XANES data demonstrates that the chemical environment of sodium is different from the surface adsorbed sodium ion, but presents some similarities to the sodium in sodium diformylamine with chemical bond between sodium and nitrogen.
- (4) The experimental results match well with the outcomes in theoretical simulation.

Revisions:

- (1) Revisions in the manuscript on page 3:

The sodium content of PCN-NaCA-2 was determined to be 6.9 wt.% by inductively coupled plasma atomic emission spectroscopy (ICP OES). For further understanding the chemical environment of sodium in the framework, PCN-NaCA framework was simulated with density functional theory (DFT) based on a linear melon structure with infinite repeating units; and in the optimized configuration,

there is interaction between the sodium ion and four pyridinic nitrogen atoms of two adjacent heptazine units (Figures 2b, S9). Nitrogen K-edge X-ray near-edge structure (XANES) measurements were thereafter used to investigate the interaction (Figure 2c). PCN shows typical π^* resonance at 399.3 eV corresponding to pyridinic N of the tri-s-triazine moiety;^{42,43} most importantly, blue shift and enhancement of the pyridinic N peak is observed, demonstrating the presence of coordination interaction and charge transfer from pyridinic nitrogen to sodium.^{44,45} Meanwhile, there is no shift for the other π^* resonances (401.2 eV, amino; 402.3 eV, graphitic),^{46,47} indicating no interaction between these nitrogen atoms and sodium. Furthermore, the chemical environment of the sodium was characterized with X-ray absorption spectrometer (Figure S10). Sodium in PCN-NaCA-2 presents a K-edge absorption profile that is different from that of NaSCN and NaSCN/PCN, but is very similar to the profile of sodium diformylamine. The XAS results are indicating that the sodium ion is strongly interacting with PCN-NaCA-2 framework via the coordination with pyridinic nitrogen, which matches well with the theoretical simulation.

References:

42. Zheng, Y. *et al.* Hydrogen evolution by a metal-free electrocatalyst. *Nat. Commun.* **5**, 3783 (2014).
43. Zhang, J.-R. *et al.* Accurate K-edge X-ray photoelectron and absorption spectra of g-C₃N₄ nanosheets by first-principles simulations and reinterpretations. *Phys. Chem. Chem. Phys.* **21**, 22819-22830 (2019).
44. Liang, Y. *et al.* Covalent Hybrid of Spinel Manganese–Cobalt Oxide and Graphene as Advanced Oxygen Reduction Electrocatalysts. *J. Am. Chem. Soc.* **134**, 3517-3523 (2012).
45. Lee, J. H. *et al.* Carbon dioxide mediated, reversible chemical hydrogen storage using a Pd nanocatalyst supported on mesoporous graphitic carbon nitride. *J. Mater. Chem. A* **2**, 9490-9495 (2014).
46. Leinweber, P. *et al.* Nitrogen K-edge XANES - an overview of reference compounds used to identify 'unknown' organic nitrogen in environmental samples. *J. Synchrotron Radiat.* **14**, 500-511 (2007).
47. Ripalda, J. M. *et al.* Correlation of x-ray absorption and x-ray photoemission spectroscopies in amorphous carbon nitride. *Phys. Rev. B* **60**, R3705-R3708 (1999).

(2) Revised Figures 2c, 2d in page 4 in the manuscript:

Figure 2. Synthesis and structure of the photocatalysts. (a) Synthesis of PCN and PCN-NaCA-*n*. (b) The optimized configurations of PCN and PCN-NaCA by DFT calculation. (c) N K-edge XANES of PCN and PCN-NaCA-2. The insets show enlarged peaks of pyridinic nitrogen and graphitic nitrogen.

(3) Figures S9 in the Supplementary Information on page S16:

Initial Structures:

Optimized Configuration:

Figure S9. Initial structures and the optimized configuration of PCN-NaCA.

(4) Figures S10 in the Supplementary Information on page S16:

Figure S10 Na K-edge X-ray absorption spectra. (a) NaSCN (20 % w.t.) / PCN prepared via Incipient wetness impregnation; (b) Sodium thiocyanate; (c) PCN-NaCA-2; (d) Sodium diformylamine.

Comments:

“- What is the spin state of the adsorbed O₂? This would be important.

- What is the spin state of any other intermediates considered?”

Responses:

Since the high-spin ground state of O₂ molecule is notoriously poorly described in DFT calculations, the free energy of O₂(g) was derived as $G_{O_2(g)} = 2 G_{H_2O(l)} - 2 G_{H_2} + 4.92 \text{ eV}$ (Please refer to the literatures listed below). Furthermore, the DMOI³ code (used in this work) is unable to calculate the energy of O₂ molecule with different spin states. In this work, all structural optimizations are thus performed based on *spin-unrestricted* DFT calculations.

Reference:

[Ref.1] Jones, R. O. & Gunnarsson, O. The density functional formalism, its applications and prospects. *Rev. Mod. Phys.* **61**, 689-746 (1989);

[Ref. 2] Jiao, Y., Zheng, Y., Jaroniec, M. & Qiao, S. Z. Origin of the electrocatalytic oxygen reduction activity of graphene-based catalysts: a roadmap to achieve the best performance. *J. Am. Chem. Soc.* **136**, 4394-4403 (2014).

Revisions:

Computational methods in Supplementary Information on page S5-S7.

Comments:

“- How were the O₂ adsorption sites determined? There are many possible adsorption sites - was a search for other possible adsorption sites performed?”

Responses:

- (1) Sodium cyanamate moiety plays an essential role for the superior performance in the photocatalytic H₂O₂ production in the experimental investigations; and this work mainly focus on the mechanism behind the superior performance initiated by sodium cyanamate moiety.
- (2) In addition to the systematic spectroscopy investigations, the theoretical simulation is employed for further understanding the role of cyanamate moiety in ORR. The sites closing to sodium and cyanamino-group (Sites 1 – 4 in Figure S32a) was thus screened for O₂ adsorption.
- (3) It was found that O₂ is unable to adsorb on sites 2 and 4, but the adsorption is feasible on sites 1 and 3 with E_{ad} (adsorption energy) of 0.32 eV and 0.39 eV, respectively (Figures S32b and S32c). O₂ adsorption on site 3 was selected as the optimum initial structure for the following calculation.

Revisions:

Figure S32 in Supplementary Information on page S31:

Figure S32. (a) Adsorption sites of O₂ on PCN, (b) and (c) atomic structures of O₂ on sites 1 and 3.

Notes : Sodium cyanamate moiety plays an essential role for the superior performance in the photocatalytic H₂O₂ production in the experimental investigations; and this work mainly focus on the mechanism behind the superior performance initiated by sodium cyanamate moiety. The theoretical simulation is employed for further understanding the role of cyanamate moiety in ORR. The sites closing to sodium and cyanamino-group (Sites 1 – 4) was thus screened for O₂ adsorption. It was found that O₂ is unable to adsorb on sites 2 and 4, but the adsorption is feasible on sites 1 and 3 with E_{ad} (adsorption

energy) of 0.32 eV and 0.39 eV, respectively. O₂ adsorption on site 3 was selected as the optimum initial structure for the following calculation.

Comments:

“- What are the charges found on individual species in these simulations? are they physical?”

If these overall simulation cells are electrically neutral, the presence of different ionic moieties still implies a significant role of charge transfer. However, the PBE functional used here (or any GGA) is known to describe charge transfer unphysically.”

Responses:

- (1) The Mulliken charge population values of ORR intermediates on PCN and PCN-NaCA based on different functional were summarized below. It was found that the charge population based on GGA-PBE and B3LYP is consistent with each other (Table S4).
- (2) It was found that on an electrically neutral cell (of both PCN and PCN-NaCA), ORR process is unlikely to happen, because the initial step of ORR (O₂ adsorption) is energetically unfavorable (Figure S33). When the cell is negatively charged with one additional electron, ORR on PCN and PCN-NaCA can proceed with surmountable barriers. This is consistent with recently published work from Prof. Qiao's group (*J. Am. Chem. Soc.* **133**, 20116 (2011)).

Revisions:

- (1) Table S4 on Supplementary Information in page 35:

Table S4. Mulliken charge population values of ORR intermediates on PCN and PCN-NaCA.

		O2	OOH	O	OH
GGA-PBE	PCN	-0.443	-0.338	-0.536	-0.276
	PCN-Na	-0.519	-0.443	-0.617	-0.247
B3LYP	PCN	-0.477	-0.374	-0.413	-0.314
	PCN-Na	-0.479	-0.477	-0.708	-0.285

Notes: For double-checking the charge transfer results, hybrid functional B3LYP was employed as well. The charge transfer between ORR intermediates and substrate based on hybrid functional

B3LYP is consistent with that based on GGA-PBE.

(2) Figure S33 in Supplementary Information on page S32:

(a) O₂ on PCN-NaCA

(b) OOH on PCN-NaCA

(c) O₂ on PCN

Figure S33. Initial structures and optimized configurations of O₂ on PCN-NaCA (a), OOH on

PCN-NaCA (b), and O₂ on PCN based on an electrically neutral cell.

Comments:

“- In particular, the drastic increase of the O₂ adsorption energy in the presence of Na⁺ / NCN⁻ remains unexplained from a fundamental point of view. Given that the O₂ molecule bonds so strongly, some significant charge transfer is likely. But this charge transfer could be entirely an artifact of the simulation - again, charge transfer is simply not described correctly by GGAs. A much more exhaustive analysis of the simulations would be necessary to clarify these issues.”

Responses:

- (1) Based on the improved configurations which are closer to the experimental results, the new DFT calculations states that the adsorption energy of O₂ on PCN-NaCA is lower than that on PCN; meanwhile, the interaction energy of O₂ on PCN-NaCA is higher than that on PCN (Table S5). Most importantly, the charge transfer between adsorbate and substrate is consistent with the interaction energy, i.e., the higher the interaction energy is, the more the charge transfer.
- (2) For double-checking the charge transfer results, hybrid functional B3LYP was employed as well. As shown in Table S4, the charge transfer between ORR intermediates and substrate based on hybrid functional B3LYP is consistent with that based on GGA-PBE.

Revisions:

Table S5 in Supplementary Information on page 35:

Table S5. Adsorption energy and interaction energy values of ORR intermediates on PCN and PCN-NaCA. All results are in unit of eV.

		O ₂	OOH	O	OH
E_{ad}	PCN	1.54	2.35	4.48	3.03
	PCN-NaCA	0.37	1.64	3.76	2.48
E_{in}	PCN	2.41	3.19	5.28	3.79
	PCN-NaCA	2.97	3.01	5.44	3.75

Comments:

“- Is 2 nm of "vacuum" really enough to electrostatically decouple different sites with strong dipolar characteristics from one another? 2 nm is not much at all. There should be some electrostatic interaction between the supercells considered.”

Responses:

Considering the suggestions, the adsorption energy values of O₂ and OOH on PCN-NaCA with different vacuum values were checked (Table S3). It was found that the variation of the adsorption energy is negligible, indicating that a vacuum with 2 nm is enough to eliminate the electrostatic interaction between the adjacent supercells.

Revisions:

Table S3 in Supplementary Information on page 33:

Table S3. Adsorption energy values of O₂ and OOH on PCN-NaCA with different vacuum values. All results are in unit of eV.

Vacuum/Å	20	25	30
O ₂	0.37	0.36	0.35
OOH	1.64	1.65	1.64

Notes: It was found that the variation of the adsorption energy is negligible, indicating that a vacuum with 2 nm is enough to eliminate the electrostatic interaction between the adjacent supercells.

Comments:

“- There is no "van der Waals interaction" in the PBE functional (or in any GGA). This is very well known. Given that the bonding here has van der Waals character at least in parts, their absence is a technical error that could invalidate the simulations altogether.”

Responses:

Thanks for the comment. Van der Waals interaction was considered in the revised manuscript. The GGA + vdW approach within Grimme scheme was adopted to describe the vdW interaction.

Revisions:

Computation methods in page 6 in Supplementary Information.

Comments:

“- Figure 8 shows “free energies”. And indeed, given that the energies of several reagents are partial pressure / concentration dependent, free energies should be used.

However, the authors say nothing regarding partial pressure of O₂ during the reaction, either in experiment or in the theory part. The correct way to couple the free energies of reference gas phases into such simulations is well known, e.g., *Physical Review B* 65 (3), 035406. Did the authors do this? If not, the numbers should be corrected and the appropriate analysis should be provided.”

Responses:

- (1) In this work, the reaction free energies were calculated based on a computational hydrogen electrode model developed by Nørskov *et al.* (*J. Phys. Chem. B* 2004, 108, 17886-17892.) The reaction free energy for elemental step was calculated as:

$$\Delta G = \Delta E + \Delta E_{\text{ZPE}} - T\Delta S + \Delta G_U + \Delta G_{\text{pH}}$$

where ΔE is the difference in the total energy, ΔE_{ZPE} and ΔS are the differences in the zero-point energy and the change of entropy, T is the temperature ($T = 298.15$ K in this work), ΔG_U and ΔG_{pH} are the contributions from the electrode potential (U) and pH value.

- (2) The CHE model defines that the chemical potential of a proton/electron in solution is equal to one half of the chemical potential of one gas-phase H₂. The gas-phase H₂O and H₂ were used as reference states. Since the high-spin ground state of O₂ molecule is notoriously poorly described in DFT calculations, the free energy of O₂(g) was derived as:

$$G_{\text{O}_2(\text{g})} = 2 G_{\text{H}_2\text{O}(\text{l})} - 2 G_{\text{H}_2} + 4.92 \text{ eV}$$

(*Rev. Mod. Phys.* **61**, 689-746 (1989); *J. Am. Chem. Soc.* **136**, 4394-4403 (2014)).

Therefore, the partial pressure of O₂ was not considered in this work.

Revisions:

Computation methods in page 5 – 7 in Supplementary Information.

Comments:

“- Without availability of the geometries used in the computations (all computational steps) the simulations will not be reproducible. Since numerous public repositories are now available to deposit such information, all pertinent input files should be made available.”

Responses:

All the optimized atomic structures and pertinent input files are uploaded as supporting materials.

REVIEWER COMMENTS

Reviewer #1 (Remarks to the Author):

Revisions were correctly made on the manuscript, and all the concerns were properly addressed in the final version.

Reviewer #2 (Remarks to the Author):

At this stage, I believe that the authors have more or less addressed most of my primary concerns. While I still recommend that the authors perform a more thorough fluence dependence analysis for their transient absorption measurements, this shortcoming is perhaps relatively minor. If the remaining reviewers are in unanimous agreement, I recommend that this work be further considered for publication in Nature Communications.

Reviewer #3 (Remarks to the Author):

The authors revised the paper carefully, addressing the critical concerns, and therefore I recommend accepting the paper.

Reviewer #4 (Remarks to the Author):

All my concerns have been addressed in the revisions. It can be accepted for publication

Reviewer #5 (Remarks to the Author):

The authors provided a very comprehensive response to several reviewers. I believe that this is commendable. I apologize to the authors for a review that focuses on a few and partial points only. Most of them concern the theory part.

1) The revised Fig 1a is much more precise (great) but the Na⁺ is still shown in a position that is inconsistent with the theoretical model. Should this depiction be consistent with the theory?

2) The revised theory details are very helpful. In particular, use of the COSMO model and of charged unit cells are key details that could be critical.

The use of charged supercells to obtain sensible results does raise a major question. Is the electrostatic model used here physically valid?

Specifically, the authors state that neutral cells will not produce stable adsorption and introducing a negative charge is needed. They cite JACS 133, 20116 (2011) as a rationale.

There is a critical technical difference between their work and JACS 133, 20116 (2011). In the present work, the authors use periodic boundary conditions. The JACS reference used cluster systems.

In cluster systems, a charge is physically well defined, using appropriate boundary conditions.

In periodic systems, there can be no formal charge. Any negative charge must be compensated by a neutralizing positive background.

This background will be all over the unit cell, in particular in the vacuum. There is no way to avoid it. DMol3, as per the 1996 technical description of its periodic implementation, places the positive charge background in the constant term of the Ewald summation.

This positive background can have serious consequences especially in surface / interface simulations, as is well documented in the plane wave literature.

What I do not know is how the positive charge background interacts with the COSMO model and if the COSMO model perhaps fixes the problem. Does it?

Delley's reference on COSMO (Ref 10 of the present SI) does not seem to show charged supercell examples (although I may have missed this point in Delley's paper).

Do the authors correct for the positive background charge? Or does the COSMO charge do this for them? It would be important to clarify this point since, to my knowledge, this type of application is not standard in the literature. Without such a clarification, the occurrence of a positive charge background in the vacuum region could have serious consequences.

3) What is the partial charge associated with Na in the overall negative calculations?

4) The authors do not include O₂ partial pressure in the computations but their experimental results now indicate a key impact of O₂ partial pressure. Can this difference be rationalized? Shouldn't the O₂ partial pressure be accounted for in the simulations?

Point-to-Point Response to Reviewers' Comments

(Manuscript NO.: NCOMMS-20-36228B)

Reviewer #1 (Remarks to the Author):

Revisions were correctly made on the manuscript, and all the concerns were properly addressed in the final version.

Reviewer #2 (Remarks to the Author):

Comments:

“At this stage, I believe that the authors have more or less addressed most of my primary concerns. While I still recommend that the authors perform a more thorough fluence dependence analysis for their transient absorption measurements, this shortcoming is perhaps relatively minor. If the remaining reviewers are in unanimous agreement, I recommend that this work be further considered for publication in Nature Communications.”

Response:

We are so grateful for the reviewer's effort on improving this work. As shown in the revised Supplementary Information (Figure S27), the data plots deviate from a quadratic scaling significantly, stating that two-photon excitation is not relevant to the power densities used in our experiments. We tend to believe that the deviation is too significant, and the additional data plots are thus not necessary.

Figure S27. The relevance of the power densities to the theoretical quadratic scaling. (a) with dioxxygen; (b) under vacuum.

Reviewer #3 (Remarks to the Author):

The authors revised the paper carefully, addressing the critical concerns, and therefore I recommend accepting the paper.

Reviewer #4 (Remarks to the Author):

All my concerns have been addressed in the revisions. It can be accepted for publication

Reviewer #5 (Remarks to the Author):

Comments:

“The authors provided a very comprehensive response to several reviewers. I believe that this is commendable. I apologize to the authors for a review that focuses on a few and partial points only. Most of them concern the theory part.”

Response:

We appreciate the reviewer’s great effort in reviewing this manuscript. The suggestions are quite valuable for improving the manuscript.

Comments:

“1) The revised Fig 1a is much more precise (great) but the Na⁺ is still shown in a position that is inconsistent with the theoretical model. Should this depiction be consistent with the theory?”

Response:

The position of sodium ion has been revised, and is now consistent with that in the theoretical model.

Revision:

Figure 2a in revised manuscript on page 4:

Figure 2. Synthesis and structure of the photocatalysts. (a) Synthesis of PCN and PCN-NaCA-n. (b) The optimized configurations of PCN and PCN-NaCA by DFT calculation. (c) N K-edge XANES of PCN and PCN-NaCA-2. The insets show enlarged peaks of pyridinic nitrogen and graphitic nitrogen.

Comments:

“2) The revised theory details are very helpful. In particular, use of the COSMO model and of charged unit cells are key details that could be critical.

The use of charged supercells to obtain sensible results does raise a major question. Is the electrostatic model used here physically valid?

Specifically, the authors state that neutral cells will not produce stable adsorption and introducing a negative charge is needed. They cite JACS 133, 20116 (2011) as a rationale.

There is a critical technical difference between their work and JACS 133, 20116 (2011). In the present work, the authors use periodic boundary conditions. The JACS reference used cluster systems.

In cluster systems, a charge is physically well defined, using appropriate

boundary conditions.

In periodic systems, there can be no formal charge. Any negative charge must be compensated by a neutralizing positive background.

This background will be all over the unit cell, in particular in the vacuum. There is no way to avoid it. DMol3, as per the 1996 technical description of its periodic implementation, places the positive charge background in the constant term of the Ewald summation.

This positive background can have serious consequences especially in surface / interface simulations, as is well documented in the plane wave literature.

What I do not know is how the positive charge background interacts with the COSMO model and if the COSMO model perhaps fixes the problem. Does it?

Delley's reference on COSMO (Ref 10 of the present SI) does not seem to show charged supercell examples (although I may have missed this point in Delley's paper).

Do the authors correct for the positive background charge? Or does the COSMO charge do this for them? It would be important to clarify this point since, to my knowledge, this type of application is not standard in the literature. Without such a clarification, the occurrence of a positive charge background in the vacuum region could have serious consequences.”

Response:

(1). We are sorry to miss a clarification on this point in the previous revision. To check the possible interaction between the positive charge background and the COSMO model, we examined the adsorption energy of OOH on PCN-NaCA with a “cell-extrapolation” method (using increasingly larger cells; ref.: *J. Phys. Chem. Lett.* **2015**, 6, 2663). As the compensating background charge is distributed homogeneously over the unit cell, the density of background charge decreases as the unit cell of PCN-NaCA increases. However, there is no obvious change in the adsorption energy of OOH on PCN-NaCA observed with the increase of the cell dimensions (Table S4). We, therefore, tend to believe that the effect of background charge on the COSMO model is negligible in this case. This clarification part has been

- added in the revised Supplementary Information. We also find similar method reported in literature (*Electrochimica Acta* **2020**, 354, 136620).
- (2). In addition, using charged unit cell and the continuum dielectric model was also found in some recent literatures, although these works were based on VASP (*ACS Catal.* **2020**, 10, 12148; *ACS Catal.* **2020**, 10, 4048).
- (3). The theoretical simulation is employed for investigating the differences in the ORR steps on PCN and PCN-NaCA; and the reaction mechanisms on PCN and PCN-NaCA are compared under the same conditions, such as the charged unit cell and the presence of neutralizing positive background. In the presence of the above-mentioned clarification, the discussions on the distinctions in the reaction mechanism on PCN and PCN-NaCA are now more reasonable in the revised manuscript.

Revisions:

- (1) Revisions in the experimental part in Supplementary Information on page S7.

To check the possible interaction between the positive charge background and the COSMO model, we examined the adsorption energy of OOH on PCN-NaCA with a “cell-extrapolation” method using increasingly larger cells;¹⁷ there is no obvious change in the adsorption energy of OOH on PCN-NaCA observed with the increase of the cell dimensions (Table S4).

Reference:

[17] Chan, K. & Nørskov, J. K. Electrochemical barriers made simple. *J. Phys. Chem. Lett.* **6**, 2663-2668 (2015).

- (2) Table S4 in Supplementary Information on page S33.

Table S4. The adsorption energy (E_{ad}) of OOH on PCN-NaCA with different cell dimensions.

lattice /Å ³	13.99×23×20	13.99×30×25	13.99×35×30	13.99×40×35
E_{ad} /eV	1.64	1.64	1.63	1.63

Comments:

“3) What is the partial charge associated with Na in the overall negative calculations?”

Response:

The Mulliken charge population of the negatively charged PCN-NaCA cell is added in the Supplementary Information (Figure S9b). The charge of Na is 0.838 e.

Revisions:

Figure S9 in Supplementary Information on page 16.

Figure S9. (a) Initial structures and the optimized configuration PCN-NaCA; (b) the Mulliken charge population of PCN-NaCA with one extra electron.

Comments:

“4) The authors do not include O₂ partial pressure in the computations but their experimental results now indicate a key impact of O₂ partial pressure. Can this difference be rationalized? Shouldn't the O₂ partial pressure be accounted for in

the simulations?"

Response:

- (1). We may not clearly explain this experiment in the previous revision. The experimental comparison of the H₂O₂ production performance under different atmosphere was for supporting that the H₂O₂ production comes from the dioxygen reduction reaction, rather than the other reactions.
- (2). The reduction reaction of dissolved oxygen (O₂ *diss.*) to H₂O₂ could be described by the following equation:

The H₂O₂ production rate is:

$$\frac{d[H_2O_2]}{dt} = k_1 [O_{2 \text{ diss.}}] [H^+]^2 [e^-]^2$$

Meanwhile, the electrons come from excitation of the photocatalyst by photons:

The electrons generation rate is:

$$\frac{d[e^-]}{dt} = k_2 F_{h\nu}$$

Therefore, H₂O₂ production rate is determined both by the concentration of dissolved oxygen and the electrons generation rate. With sufficient photo-induced electrons supply, H₂O₂ production rate is determined by the concentration of dissolved oxygen; and H₂O₂ production rate will be controlled by electrons generation rate when the oxygen supply is in excess, which is desirable for mechanistic studies on the photocatalytic system.

In the experimental part, the concentration of dissolved oxygen in 1 atm O₂ atmosphere is roughly 5 times of that in air atmosphere; however, the H₂O₂ production rate of the reaction system under 1 atm O₂ is only 1.6 times

of that in air atmosphere, which states that the dissolved O_2 is an excess reagent under the reaction conditions in this project.

The impact of the oxygen partial pressure on the H_2O_2 production performance in the experimental part could be understood via the discussion on the reaction kinetics. We made revision in the manuscript on this point.

Revision:

Revision in manuscript on page 5.

As dioxygen is the source for H_2O_2 production, the oxygen partial pressure directly impacts the H_2O_2 production performance.

REVIEWERS' COMMENTS

Reviewer #5 (Remarks to the Author):

I reread the manuscript and thank the reviewers for their answers and additions. The work presented here is still impressively comprehensive and should be accepted.

The supercell size dimension test (Table S4) hopefully settles the potential problem of a charged supercell in practice.

Regarding the O₂ partial pressure. What the authors are doing in the experimental part is consistent with what I meant to say for the theory part.

Specifically, the O₂ partial pressure, if considered, would enter the calculation as an O₂ chemical potential (dependent on temperature and pressure for instance via ideal gas laws). This would show up as a pressure/temperature dependent ideal gas term in ΔS for O₂, on page S6, line 180.

My understanding is now that the authors likely did not consider this term. In any event, I suspect that this term would only shift the energy zero in Figure 8, relative to all other terms.

If that is so, a short comment might be good (and all that is needed).

Very minor comment: I think some text segments (also in the SI) can still benefit from some polishing for correct English.

I did not pay close attention, but, e.g.:

- line 198 "exited photocatalyst" (excited?)
- line 223 This sentence ("As the non-emissive ...") is neither complete nor grammatically correct
- Figure 6, caption: Is this really an "Excitation [sic] Energy"? Isn't this the photon fluence?
- similar in line 238
- line 282 "photons excitation" -> "photon excitation"
- line 349 "and prolongs their lifetime" is in the wrong place
- line 351 "taking" -> "taken"
- SI line 179 incorrect singular vs. plural in that sentence

[there are more missing articles etc. throughout the text]

Other typo (line 162): "intermediates"

Point-to-Point Response to Reviewers' Comments
(Manuscript NO.: NCOMMS-20-36228B)

Reviewer #5 (Remarks to the Author):

Comments:

I reread the manuscript and thank the reviewers for their answers and additions. The work presented here is still impressively comprehensive and should be accepted.

--

The supercell size dimension test (Table S4) hopefully settles the potential problem of a charged supercell in practice.

--

Regarding the O₂ partial pressure. What the authors are doing in the experimental part is consistent with what I meant to say for the theory part.

Specifically, the O₂ partial pressure, if considered, would enter the calculation as an O₂ chemical potential (dependent on temperature and pressure for instance via ideal gas laws). This would show up as a pressure/temperature dependent ideal gas term in Delta S for O₂, on page S6, line 180.

My understanding is now that the authors likely did not consider this term. In any event, I suspect that this term would only shift the energy zero in Figure 8, relative to all other terms.

If that is so, a short comment might be good (and all that is needed).

Very minor comment: I think some text segments (also in the SI) can still benefit from some polishing for correct English.

I did not pay close attention, but, e.g.:

- line 198 "exited photocatalyst" (excited?)

- line 223 This sentence ("As the non-emissive ...") is neither complete nor grammatically correct

- Figure 6, caption: Is this really an "Excitation [sic] Energy"? Isn't this the photon fluence?

- similar in line 238
- line 282 "photons excitation" -> "photon excitation"
- line 349 "and prolongs their lifetime" is in the wrong place
- line 351 "taking" -> "taken"
- SI line 179 incorrect singular vs. plural in that sentence
[there are more missing articles etc. throughout the text]
- Other typo (line 162): "intermediates"

Response:

We are grateful to the reviewer's effort in improving this manuscript.

We have double checked the manuscript and all the typo and grammatical error have been corrected, which are highlighted in blue in the revised manuscript.